# MEMORIZATION IN IN-CONTEXT LEARNING

## ABSTRACT

In-context learning (ICL) has proven to be an effective strategy for improving the performance of large language models (LLMs) with no additional training. However, the exact mechanism behind this performance improvement remains unclear. This study is the first to show how ICL surfaces memorized training data and to explore the correlation between this memorization and performance on downstream tasks across various ICL regimes: zero-shot, few-shot, and many-shot. Our most notable findings include: (1) ICL significantly surfaces memorization compared to zero-shot learning in most cases; (2) demonstrations, without their labels, are the most effective element in surfacing memorization; (3) ICL improves performance when the surfaced memorization in few-shot regimes reaches a high level (about 40%); and (4) there is a very strong correlation between performance and memorization in ICL when it outperforms zero-shot learning. Overall, our study uncovers memorization as a new factor impacting ICL, raising an important question: to what extent do LLMs truly generalize from demonstrations in ICL, and how much of their success is due to memorization?

## 1 INTRODUCTION

In-context learning (ICL) has emerged as a powerful method for improving the performance of large language models (LLMs) without extra training (Brown et al., 2020). This method involves including a few task-specific examples, known as demonstrations or shots, within the input prompt, which enables the LLM to infer the target task and generate improved responses. With long-context LLMs (OpenAI, 2023; Anil et al., 2023; Lu, 2023, inter alia), ICL has evolved to incorporate hundreds or even thousands of demonstrations, leading to greater performance improvements (Bertsch et al., 2024; Agarwal et al., 2024; Zhang et al., 2023b). However, despite its widespread use and straightforward nature, the underlying principles of ICL and its performance improvement capabilities remain unclear (Min et al., 2022b; von Oswald et al., 2023; Razeghi et al., 2022, inter alia).

In this work, we further study the inner workings of ICL by investigating the previously unexplored relationship between *ICL* and *memorization* of training data in LLMs, and how this memorization correlates with performance. In particular, to show how ICL surfaces memorization, we replace the *learning component* (target variable) in ICL with a *text completion task* which is based solely on *memorization*. To achieve this, we adapt the data contamination detection method proposed by Golchin & Surdeanu (2023a). This method aims to replicate dataset instances through *memorization* to verify their presence in the training data. The process begins by splitting a dataset instance into two random-length segments. The initial segment and the corresponding label of the dataset instance are integrated into the input prompt, instructing the LLM to generate the subsequent segment. The generated completion is then evaluated against the original subsequent segment and categorized as an exact, near-exact, or inexact match, with the first two indicating memorization. To implement this for ICL, we use the same strategy to replicate dataset instances, but with a tweak: we include a few pairs of initial and subsequent segments from different dataset instances, along with their labels in the input prompt, as *demonstrations*. Specifically, each demonstration consists of (1) a pair of initial and subsequent segments, and (2) a label. We then quantify the memorization across various regimes (i.e., zero-shot, few-shot, and many-shot) by counting the number of exact and near-exact matches. Figure 1 shows prompts for an illustrative two-shot ICL to replicate a dataset instance.

We examine memorization across various $k$-shot scenarios, where $k = \{0, 25, 50, 100, 200\}$. Here, demonstrations for smaller $k$ values are subsets of those used for larger $k$ values. We categorize our experiments into three *regimes* based on $k$ values: zero-shot for $k = 0$, few-shot for $k = \{25, 50\}$,

Figure 1: **Illustrative examples of a two-shot ICL prompt for replicating instances from NLI (left) and classification (right) tasks.** Note that, in our actual experiments, we use $k$-shot ICL, where $k = \{0, 25, 50, 100, 200\}$. All colored segments, except the green one, form the input prompt. Specifically, the gray segments indicate the instruction, the red segments display the two demonstrations, the blue segments correspond to the dataset instance being replicated, and the green segment exhibits the generated completion by the underlying LLM (GPT-4) for the subsequent segment of the dataset instance being replicated. For both examples, the generated completions are exact matches.

and many-shot for $k = \{100, 200\}$. Each regime is analyzed under different *settings* to identify the key element contributing the most to memorization by ICL. These elements include (1) instruction, (2) segment pairs, and (3) their respective labels, with the latter two forming the demonstrations. We vary the amount of in-context information in each setting by selectively including/excluding these elements in the prompt to identify the impact of each element on memorization. First, all elements are included—an instruction containing dataset-specific information (i.e., dataset and partition name) and segment pairs with their labels—to establish an upper bound for memorization. Figure 1 depicts this setting. Second, we remove the instruction (gray parts in Figure 1). Third, we exclude the instruction and labels, leaving only segment pairs (red parts in Figure 1, without labels). Finally, we evaluate the correlation between surfaced memorization and performance in all settings.

The primary contributions of this paper are as follows:

1. For the first time, we study the relationship between ICL and memorization in LLMs.

2. Our study identifies the key element contributing to surfacing memorization by ICL.

3. We explore the correlation between memorization and performance in ICL.

4. By analyzing the surfaced memorization levels in ICL, we identify cases where ICL either succeeds or fails to outperform zero-shot learning.

We made several important observations, summarized in the key findings below:[1]

1. ICL with only a few demonstrations (e.g., 25 shots) surfaces significant memorization in most cases for data that is part of the training set.

2. *Segment pairs—demonstrations without their labels—*are the key element contributing to surfacing memorization by ICL.

3. *There is a very strong correlation between performance on downstream tasks and surfaced memorization by ICL* when it improves performance compared to zero-shot learning.

4. ICL outperforms zero-shot learning when the surfaced memorization in few-shot regimes is significant, specifically when the memorization level is about 40% or higher.

5. As demonstrations increase, even though the surfaced memorization by ICL remains relatively constant at a high level in most many-shot regimes, near-exact matches gradually become exact matches, making memorization more explicit.

---

[1]See Section 5 for a comprehensive list of observations.

6. Evaluating performance on memorized and non-memorized instances in ICL reveals that the performance on memorized instances is consistently higher than non-memorized instances across nearly all regimes, from zero-shot to many-shot.

7. Consistent with the findings of Carlini et al. (2023) on memorization in language models, we discovered that memorization significantly increases with the number of tokens of context used to prompt the model. However, our experiments further this finding by showing that these *tokens can be from individual instances*, not only tokens from a single instance.

## 2  TERMINOLOGY

Before discussing our methodology, we establish specific terminology for clarity and consistency.

**Element:** We use the term "element" to refer to any of the following: instruction, segment pairs, or labels. In our experiments, we assess the impact of each element on memorization by ICL.

**Setting:** One of the key objectives of this study is to identify the main element influencing memorization in ICL. For this, we experiment with three settings, each varying by the amount of in-context information in the input prompt. Thus, the term "setting" refers to *the amount of information incorporated into the input prompt* in our experiments. We detail our settings in Subsection 3.2.

**Regime:** Contrary to settings, we define regimes based on the values of $k$ in $k$-shot scenarios. Hence, the term "regime" emphasizes the *number of demonstrations (shots) used in the input prompt*. We elaborate on these regimes in Subsection 3.4.

**Demonstration:** In the scope of ICL, several terms describe task-specific examples included in the input prompt. For clarity, we use the terms "demonstrations" and "shots" interchangeably to refer to these examples. As previously noted, each demonstration in our experiments comprises (1) a segment pair with an initial and subsequent segment, and (2) a label. *Therefore, when we mention demonstrations without labels, we only refer to segment pairs.*

## 3  APPROACH

### 3.1  DETECTING AND QUANTIFYING MEMORIZATION

To detect and quantify memorization by ICL, we use the method proposed by Golchin & Surdeanu (2023a), originally designed to detect data contamination in LLMs. Below, we explain how we adjust the original method to detect memorization by ICL and detail the procedure for quantifying it.

**Detecting Memorization in In-Context Learning.** We specifically employ the "guided instruction" strategy from Golchin & Surdeanu (2023a). This approach aims to verify if specific instances from a particular dataset partition (e.g., test set) were included in the model's training data by replicating them through memorization. To this end, each dataset instance is split into two random-length segments, and the LLM is then tasked with completing the subsequent segment based on the initial segment and the respective label provided in the input prompt. The prompt also incorporates dataset-specific details (i.e., dataset and partition name) to better guide the LLM in the replication process.

To adapt this strategy for $k$-shot ICL, we include $k$ pairs of initial and subsequent segments from $k$ distinct dataset instances, along with their labels, as *demonstrations* in the input prompt. With this prompt, we follow the same process of replicating the subsequent segments for the dataset instances under consideration. Finally, as in Golchin & Surdeanu (2023a), the similarity between the generated completions and the original subsequent segments is evaluated to determine if the dataset instances were part of the model's training data. Figure 1 provides examples of demonstrations and their integration into our replication process to study memorization by ICL.

**Evaluating Memorization in In-Context Learning.** Golchin & Surdeanu (2023a) proposed three categories for evaluating generated completions against the original subsequent segments:

**(1) Exact Match:** The completion exactly matches the original subsequent segment.

**(2) Near-Exact Match:** The completion, while not identical, shows considerable overlap and maintains significant semantic and structural similarity to the original subsequent segment.[2]

**(3) Inexact Match:** The completion is completely different from the original subsequent segment.

They employed GPT-4 with few-shot ICL to classify generated completions into these categories. In particular, this classifier uses a few human-annotated examples of exact and near-exact matches in the prompt as references and automatically compares the generated completions to their original counterparts.[3] We adopt the same method and adhere to the same categories for our evaluation. Although their results showed this evaluation strategy achieves high accuracy (92–100%) in matching evaluations from human judgments, we conduct an additional human evaluation on top of GPT-4's evaluation to ensure optimal accuracy in our findings. This is important as our conclusions significantly rely on the number of detected exact and near-exact matches. We detail our human evaluation process in Section 4, under Human Evaluation.

**Quantifying Memorization in In-Context Learning.** Following Golchin & Surdeanu (2023a), we consider both exact and near-exact matches as indicators of memorization. We quantify memorization by counting the number of these matches and expressing them as a percentage of the total dataset instances under consideration.

### 3.2 IDENTIFYING THE KEY ELEMENT IN MEMORIZATION IN IN-CONTEXT LEARNING

Our experiments involve three distinct settings, all aiming at quantifying memorization but differing in the amount of information included in the input prompt. This helps us measure the amount of memorization in ICL regimes based on the information provided by each element and identify the key element in the process. As shown in Figure 1 and discussed in Section 2, the input prompt is composed of two main parts: the *instruction*, which contains dataset-specific details, and *demonstrations*, which include *segment pairs* and their respective *labels*. We combine these three elements—*instruction*, *segment pairs*, and *labels*—in different ways to create three unique settings with varying amounts of in-context information. Below, we detail each setting.

**(1) Full Information.** This setting maximizes in-context information by *including all three elements: instruction, segment pairs, and labels*. Figure 1 illustrates this setting. In fact, this setting contains more information than standard ICL by incorporating dataset-specific details not typically included. We use it to establish an upper bound for the highest possible amount of memorization that can be surfaced in ICL regimes. By comparing the impact of each element on memorization against this maximum, we identify which element most significantly influences memorization in ICL.

**(2) Segment Pairs and Labels.** Here, we exclude the instruction containing dataset-specific information and *include only segment pairs and labels*. To show this setting, it omits the gray segments in Figure 1 and includes only the red segments. This setting is closest to standard ICL, although standard ICL includes an instruction for executing the target task, which is absent here. However, since this instruction lacks relevant information that can affect memorization, its impact on memorization is zero or negligible. Additionally, as we see in Section 5, even an instruction with dataset-specific information (as in the previous setting) has minimal impact on memorization in ICL regimes.

**(3) Only Segment Pairs.** We further remove elements from the input prompt and *include only segment pairs*, excluding the instruction and labels. While the previous setting examines the combined effect of segment pairs and labels on memorization in ICL, this setting shows their individual contributions. By comparing the amount of surfaced memorization in this setting with the one that includes both segment pairs and labels, as well as the full information setting, we can assess how much memorization is due to segment pairs alone versus labels. This helps identify the primary element driving memorization across ICL regimes.

### 3.3 PERFORMANCE AND MEMORIZATION IN IN-CONTEXT LEARNING

As the primary goal of using ICL is to enhance downstream performance, we explore the connection between memorization by ICL and performance. We compute the performance on the samples for which we assess memorization and analyze the correlation between performance and memorization

---

[2]Examples of exact and near-exact matches are shown in Table 3 in Appendix A.

[3]Figure 5 in Appendix B illustrates this evaluation prompt.

across our three settings using the Pearson correlation (Pearson, 1895). In addition, we separately evaluate performance for *memorized* and *non-memorized* instances across ICL regimes to further explore this relationship. According to Subsection 3.1, instances that are replicated exactly or nearly exactly are considered memorized, while those replicated inexactly are considered non-memorized. Note that, for performance measurement, we use standard $k$-shot ICL, which includes an instruction to perform the task with $k$ demonstrations and their labels embedded in the input prompt.

### 3.4 SELECTION OF IN-CONTEXT LEARNING REGIMES

We work with five $k$-shot ICL across our three settings, where $k = \{0, 25, 50, 100, 200\}$, covering all ICL regimes: zero-shot, few-shot, and many-shot. Specifically, we define zero-shot regimes when $k = 0$, few-shot regimes when $k = \{25, 50\}$, and many-shot regimes when $k = \{100, 200\}$. In our experiments, to assess the impact of increasing demonstrations on memorization and performance, we progressively increase the number of demonstrations, ensuring that larger regimes include all demonstrations from the smaller ones. For example, the 100-shot ICL includes 50 demonstrations from the 50-shot ICL, which itself includes 25 demonstrations from the 25-shot ICL.

### 3.5 SELECTION OF MODELS

To achieve the goals of our study, the LLMs must meet specific criteria to be selected. First, they must be highly performant, with strong steerability and controlled generation capabilities, enabling us to effectively quantify memorization through their outputs. This is crucial given the opaque nature of the training data—if a model fails to replicate a dataset instance, we can reasonably conclude it was not part of the training data, rather than attributing it to the model's inability to replicate. Less performant models may keep memorization internal by not explicitly emitting memorized data, or generate unstructured outputs that make detecting memorized data intangible. Second, as we extend our experiments to many-shot regimes, the LLMs must support long contexts to accommodate our largest many-shot regime with 200 demonstrations across all datasets. Third, the candidate LLMs must have been trained on an array of datasets. This diversity is key for observing how memorization evolves across different ICL regimes through instance replication. Clearly, without this criterion, studying memorization is unfeasible. Note that, if an LLM does not meet these criteria, it does not invalidate our conclusions. In fact, these criteria are essential for effectively *studying* memorization, but memorization exists in all language models regardless (Carlini et al., 2023; 2021).

### 3.6 SELECTION OF DATASETS

In line with the settings outlined in Subsection 3.2, the datasets for our study must fulfill certain criteria. First, the datasets must be part of the training corpora for the LLMs used in our study, ensuring that their instances can be potentially replicated through memorization. Second, to evaluate the impact of labels on memorization in ICL regimes, we need datasets with labeled samples. Third, these datasets should have a complex label space or be challenging enough for LLMs, allowing us to observe performance change across ICL regimes and explore its correlation with memorization. Fourth, the sample length must be limited to a few dozen tokens to fit within the input context length of LLMs for all datasets, handling up to 200 demonstrations in our largest many-shot regime.

## 4 EXPERIMENTAL SETUP

**Model.** Per the criteria detailed in Subsection 3.5 for selecting models, we conducted a pilot study to determine which existing LLMs fulfill all requirements. We initially selected a set of long-context, high-performing LLMs, including GPT-4 (OpenAI, 2023), GPT-4o (OpenAI, 2023), Gemini 1.5 Pro (Anil et al., 2023; Reid et al., 2024), and Claude 3.5 Sonnet (Anthropic, 2024). Our pilot study found that GPT-4o and Gemini 1.5 Pro struggled with controlled generations, particularly in many-shot regimes, and Claude 3.5 Sonnet was unable to perform our tasks due to strict safety filters preventing the generation of copyrighted content—in our case, replicating dataset instances. Among the models tested, only GPT-4 showed the ability to produce controlled outputs.[4] Also, GPT-4 auto-

---

[4]Even if other LLMs met our criteria, we were restricted to one model due to the prohibitive cost of proprietary LLMs and the substantial GPU demands for running open-weight LLMs in long-context settings.

matically met the final criterion, as it was shown to have been trained on multiple datasets (Golchin & Surdeanu, 2023a). Thus, we selected GPT-4 with 32k context length for all our experiments.

In our experiments, we use GPT-4 for three different tasks: measuring memorization, computing performance, and evaluating generated completions as exact, near-exact, or inexact matches. For all these tasks, we access GPT-4 through the Azure OpenAI API. Specifically, we use the `gpt-4-0613-32k` snapshot for the first two tasks and the `gpt-4-0613` snapshot for evaluation. To promote deterministic generations, we set the temperature to zero in all our experiments. We limit the maximum completion lengths to 100 tokens for measuring memorization and 10 tokens for both computing performance and evaluating generated completions. For performance assessment, we repeat our experiments three times and report the average results.

**Data.** Based on the criteria listed in Subsection 3.6 for selecting datasets, we use four label-based datasets from two tasks: natural language inference (NLI) and classification. Although all criteria for selecting datasets can be independently verified, the first criterion must be verified in relation to the selected model—here, GPT-4. To confirm that the datasets were part of GPT-4's training data, we conducted a pilot study using proposed strategies for detecting data contamination in fully black-box LLMs (Golchin & Surdeanu, 2023a;b). Based on the results, we selected the following datasets: WNLI (Wang et al., 2019b), RTE (Wang et al., 2019a), TREC (Hovy et al., 2001b), and DBpedia (Wang et al., 2020).[5] The first two datasets are for NLI, while the latter two are for classification. Consistent with prior work (Golchin & Surdeanu, 2023a;b; Bertsch et al., 2024; Zhao et al., 2021; Lu et al., 2022; Han et al., 2023; Ratner et al., 2023), to control costs and work with a manageable sample size, we subsample 200 instances from the train split of each dataset, evenly distributed by labels, to study both memorization and performance in all our experiments. For measuring memorization, we create pairs of random-length segments for dataset instances by randomly deriving the initial segment from 60% to 80% of each instance's length, based on the white space count.

**Demonstrations.** Our preparation process for demonstrations closely follows the method used for dataset instances. Specifically, we subsample 200 demonstrations from each dataset's train set, ensuring an even label distribution to prevent majority label bias in ICL (Zhao et al., 2021). These demonstrations are then split into two random-length segments, with the initial segment containing 60% to 80% of the instance's length, based on the white space count. Finally, these 200 segment pairs along with their respective labels constitute our 200-shot ICL.

Regarding the order of demonstrations, while the order matters in few-shot regimes (Lu et al., 2022), its impact diminishes in many-shot regimes (Bertsch et al., 2024). To reduce this effect in few-shot regimes and ensure our findings are order-independent, we present demonstrations in random order within the input prompt across all experiments. However, when studying memorization and performance for the same $k$-shot ICL, the order remains unchanged. For example, in a 25-shot ICL, the order of demonstrations is random but consistent when examining memorization and performance.

**Human Evaluation.** Golchin & Surdeanu (2023a) proposed a classifier based on GPT-4 with few-shot ICL to evaluate generated completions, achieving high accuracy (92–100%) in matching human judgments. To improve upon this, we add an extra human evaluation layer. This step is beneficial, as our findings heavily rely on the number of exact and near-exact matches detected in the replication process. This ensures no tolerance for mislabeled completions. After human evaluation, only 5% of the labeled completions by GPT-4 were adjusted, all of which were borderline cases between near-exact and inexact matches. This performance aligns with the reported accuracy range.

## 5 RESULTS AND DISCUSSION

Below, we first discuss our results on memorization across various ICL regimes in our three settings. We then explore the correlation between this memorization and performance on downstream tasks.[6]

### 5.1 RESULTS ON QUANTIFYING MEMORIZATION

Figures 2 presents a series of plots that quantify memorization across various ICL regimes in our three settings: (1) full information, (2) segment pairs and labels, and (3) only segment pairs. In

---

[5]More details on datasets can be found in Appendix C.

[6]In Appendix E, we compare our observations with prior studies showcasing specific characteristics of ICL. Additionally, we discuss the practical implications of our observations in Appendix D.

this figure, memorization is represented using both exact and near-exact matches (left-hand plots), as well as exact matches alone (right-hand plots). *Unless otherwise stated, all our observations are based on memorization quantified by both exact and near-exact matches.* Further, we use the first setting only for comparing with maximum memorization, the second setting when discussing memorization by ICL (as it closely resembles standard ICL), and the third setting to identify the key element contributing the most to memorization by ICL.

We draw six observations on memorization by ICL:

**Observation 1:** ICL significantly surfaces memorization compared to zero-shot learning. For instance, in the left plot under the segment pairs and labels setting in Figure 2, memorization in zero-shot regimes ranges from 11–16.50%. This increases to 18–63.50% in few-shot regimes and further to 24–75% in many-shot, more than doubling the amount in zero-shot.

**Observation 2:** In terms of memorization behavior, providing only a few demonstrations (e.g., 25 shots) *sharply* increases memorization in ICL for most datasets. While this increase continues for larger shots in some datasets, it plateaus for others. For example, in the aforementioned plot, for the WNLI and RTE datasets, memorization increases up to 75% and 24% at 200-shot. However, for the TREC and DBpedia datasets, it levels of at around 40% and 53%, respectively, after 25-shot.

**Observation 3:** Memorization tends to remain stable across many-shot regimes for most datasets. However, within these regimes, memorization becomes more explicit as near-exact matches gradually transform into exact matches, as shown in Figure 4 in Appendix A. This highlights the importance of near-exact matches in quantifying memorization, as they indeed indicate memorization. Table 3 in Appendix A also provides examples of near-exact matches turning into exact matches as the number of demonstrations increases. In addition, as shown in all plots of Figure 2, the memorization pattern remains consistent whether quantified by exact and near-exact matches together or by exact matches alone, further validating that near-exact matches signal memorization.

**Observation 4:** As shown in the left-hand plots of Figure 2, the amount of surfaced memorization in the setting with segment pairs and labels reaches its maximum—equivalent to the full information setting—as soon as a few demonstrations (25 shots and more) are provided in the input prompt. In other words, the key difference between the full information setting and the setting with segment pairs and labels lies in the zero-shot regimes. In fact, the *dataset-specific information is overshadowed by the information from segment pairs and labels (or demonstrations) in the input prompt in terms of contributing to the memorization by ICL.*

**Observation 5:** Individual instances of the same context can significantly increase memorization. This is supported by viewing demonstrations as individual dataset instances within the same dataset (context) that contribute to memorization, as seen in all plots of Figure 2. This complements the findings of Carlini et al. (2023) that memorization significantly increases with the tokens of context used to prompt the model.[7]

**Observation 6:** Comparing memorization levels across three settings (all left-hand plots of Figure 2) shows that maximum memorization by ICL can be achieved even when *only segment pairs* are included in the input prompt. *This indicates that demonstrations without their respective labels, i.e., the segment pairs alone, are the key element contributing to memorization by ICL.*

To clarify, the WNLI dataset is not an exception to this observation, although it experiences a slightly larger decrease (e.g., 22.50% in 200-shot) compared to other datasets. In fact, we discovered that in the WNLI dataset, multiple sentence pairs share sentence 1 with different labels. For instance, for *"Bill passed the gameboy to John because his turn was next."* as sentence 1, there are different options for sentence 2, such as *"John's turn was next."* and *"Bill's turn was next."*, with distinct labels—*entailment* and *not entailment*, respectively. Hence, when the model is prompted with only sentence 1 to generate sentence 2, the completion could be either of these options. However, our criteria for exact and near-exact matches require both semantic and structural similarity. If the completion does not semantically or structurally match the original sentence 2, even if it matches the other sentence 2 with a different label, it is not counted as an exact/near-exact match. This approach is consistent with previous work on memorization in language models (Carlini et al., 2023). We found several cases of this occurring in the WNLI dataset, and the number of such cases exactly corresponds with the drop seen for the WNLI dataset in the setting containing only segment pairs.

---

[7]See Appendix E for further discussion.

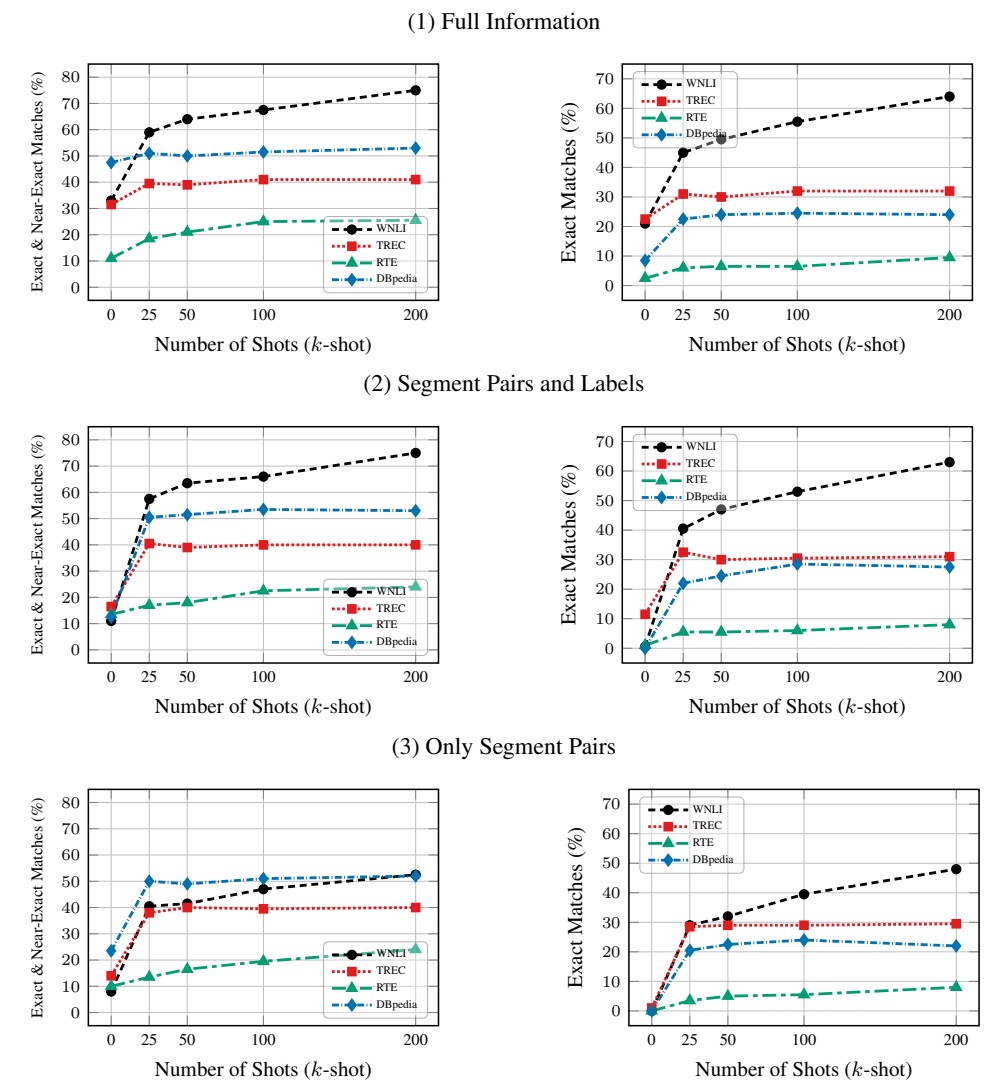

Figure 2: **Results on quantifying memorization in different ICL regimes for all settings.** Plots on the left display memorization quantification using *exact and near-exact matches* and plots on the right illustrate this using *only exact matches*. GPT-4 is the underlying model in all settings.

Table 1: Pearson correlation between overall performance and memorization across all settings. Here, memorization is quantified using *both exact and near-exact matches.*

| Setting | WNLI | TREC | DBpedia | RTE |
|---|---|---|---|---|
| Full Information | 0.98 | 0.95 | 0.91 | –0.55 |
| Seg. Pairs & Labels | 1.00 | 0.91 | 0.88 | –0.30 |
| Only Seg. Pairs | 0.99 | 0.92 | 0.89 | –0.30 |

Table 2: Pearson correlation between overall performance and memorization across all settings. Here, memorization is quantified using *only exact matches.*

| Setting | WNLI | TREC | DBpedia | RTE |
|---|---|---|---|---|
| Full Information | 0.97 | 0.87 | 0.88 | –0.40 |
| Seg. Pairs & Labels | 0.99 | 0.77 | 0.93 | –0.50 |
| Only Seg. Pairs | 0.97 | 0.82 | 0.89 | –0.47 |

## 5.2 RESULTS ON PERFORMANCE AND MEMORIZATION

Figure 3 shows a series of plots comparing performance (left-hand plots) and memorization (right-hand plots) across different ICL regimes in all our three settings. Accordingly, Tables 1 and 2 list the Pearson correlation coefficients between overall performance and memorization, with memorization quantified using both exact and near-exact matches, and only exact matches, respectively.

We list three observations about the connection between performance and memorization in ICL:[8]

**Observation 1:** ICL outperforms zero-shot learning when the surfaced memorization level in few-shot regimes is substantial, reaching around 40% or higher. This is evident in the results from the WNLI, TREC, and DBpedia datasets in Figure 3.

**Observation 2:** Performance on memorized instances is consistently higher than on non-memorized instances across nearly all settings, from zero-shot to many-shot regimes. This can be observed in the left-hand plots of Figure 3.

**Observation 3:** As indicated in Tables 1 and 2, *when providing demonstrations in ICL leads to performance improvement compared to zero-shot learning, this is highly correlated with memorization.* Specifically, Pearson coefficients provide very strong evidence of this correlation.

# 6 RELATED WORK

**In-Context Learning.** Proposed by Brown et al. (2020), ICL enhances performance in LLMs without additional training by including a few demonstrations in the input prompt. Despite its simplicity, the internal mechanism of ICL is not yet well understood. Several studies explored ICL from various perspectives: Lu et al. (2022) examined the impact of the order of demonstrations, Bölücü et al. (2023) investigated the effect of example selection, Zhao et al. (2021) explored label, recency, and common token biases in ICL, Li et al. (2023b) studied the influence of input distribution and explanations, and Min et al. (2022b) looked into the role of labels and found that randomly replacing labels does not harm performance while others showed that this is not true for all tasks and models (Yoo et al., 2022; Kossen et al., 2023; Lin & Lee, 2024). Different perspectives were employed to better understand ICL: Hendel et al. (2023) viewed ICL as compressing the training set into a single task vector to produce output, while von Oswald et al. (2023) interpreted ICL as gradient descent, a view refuted by Deutch et al. (2023). Some research efforts focused on maximizing ICL performance through different paradigms: several studies extremely increased demonstrations in the input prompt (Bertsch et al., 2024; Agarwal et al., 2024; Zhang et al., 2023b; Milios et al., 2023; Anil et al., 2023), Min et al. (2022a) fine-tuned models to perform ICL, and Zhao et al. (2021) used prompt engineering to enhance ICL. Recent research also revealed additional capabilities of ICL beyond its performance on standard benchmarks, such as performing regression (Vacareanu et al., 2024), kNN (Agarwal et al., 2024; Dinh et al., 2022), and jailbreaking (Anil et al., 2024).

**Memorization.** Memorization is a well-studied topic in the context of language models (Carlini et al., 2023; 2021; Song & Shmatikov, 2019; Zhang et al., 2023a; McCoy et al., 2023; Song et al., 2017). While some studies aimed to verify the presence of memorization (Henderson et al., 2018; Thakkar et al., 2020; Thomas et al., 2020; Carlini et al., 2019; Golchin & Surdeanu, 2023b), others attempt to quantify it (Carlini et al., 2023; 2021). Quantifying memorization is typically handled using membership inference attack (Shokri et al., 2017; Yeom et al., 2018) to reproduce training data from models (Carlini et al., 2023; 2021; Golchin & Surdeanu, 2023a). This quantification is conducted for several reasons, including showcasing potential privacy risks (Nasr et al., 2023; Biderman et al., 2023; Lukas et al., 2023), addressing copyright infringement (Grynbaum & Mac, 2023; Karamolegkou et al., 2023), and generating factual information (Li et al., 2023a; Tay et al., 2022; AlKhamissi et al., 2022; Petroni et al., 2019; Haviv et al., 2023). On the other hand, several works proposed methodologies to mitigate memorization (Lee et al., 2022) and its implications (Kandpal et al., 2022; Meeus et al., 2024; Wei et al., 2024; Wang et al., 2023).

To our knowledge, no prior work studied memorization in ICL and its correlation with performance.

# 7 CONCLUSION

We studied memorization in in-context learning (ICL) and its correlation with downstream performance in large language models (LLMs). We quantified this memorization and identified the most effective element in surfacing it. Our key findings are: (1) ICL significantly surfaces memorization compared to zero-shot learning in most cases, with demonstrations—excluding their labels—being the most influential element; and (2) there is a very strong correlation between performance and

---

[8]See Appendix F for an extended discussion.

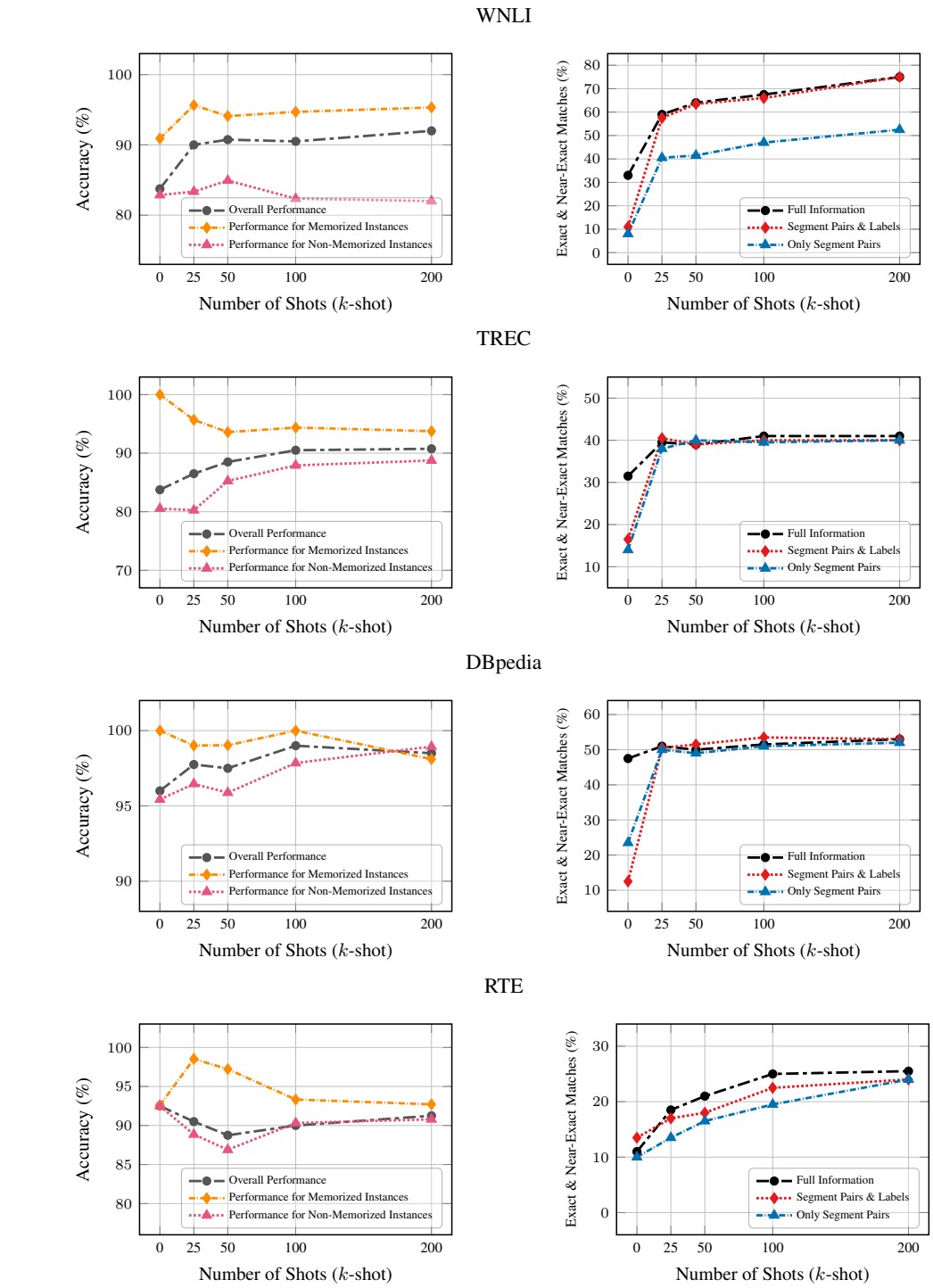

Figure 3: **Performance vs. memorization in different ICL regimes for all settings.** Plots on the left show performance and plots on the right display memorization across all three settings in one view. Note that the memorization plots are duplicated from Figure 2 for comparison purposes.

memorization when ICL outperforms zero-shot learning. Overall, our findings highlight memorization as a new factor impacting ICL. While our research offers a deeper understanding of ICL, it also presents new challenges, particularly regarding how much LLMs truly learn from demonstrations in ICL versus how much of their success is due to memorization.

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

Table 3: **Examples of near-exact matches becoming exact matches across various ICL regimes.** Each prompt indicates the number of demonstrations (shots) with a $\{k\ shots\}$ placeholder to save space. For clarity, examples are provided in the full information setting. In WNLI and DBpedia datasets, near-exact matches from 25-shot become exact matches at 50-shot. Similarly, in the TREC dataset, the transition from near-exact to exact match occurs when moving from 25-shot to 100-shot.

| Near-Exact Match | Exact Match |
|---|---|
| **Instruction:** You are provided with Sentence 1 from the train split of the WNLI dataset. Finish Sentence 2 as appeared in the dataset. Sentence 2 must exactly match the instance in the dataset. 
 − − − 
 *{25 shots}* 
 − − − 
 **Label:** 1 (entailment) 
 **Sentence 1:** Steve follows Fred's example in everything. He influences him hugely. 
 **Sentence 2:** 
 `Fred influences Steve hugely.` | **Instruction:** You are provided with Sentence 1 from the train split of the WNLI dataset. Finish Sentence 2 as appeared in the dataset. Sentence 2 must exactly match the instance in the dataset. 
 − − − 
 *{50 shots}* 
 − − − 
 **Label:** 1 (entailment) 
 **Sentence 1:** Steve follows Fred's example in everything. He influences him hugely. 
 **Sentence 2:** 
 `Fred influences him hugely.` |
| **Instruction:** You are provided with the first piece of an instance from the train split of the DBpedia dataset. Finish the second piece of the instance as exactly appeared in the dataset. 
 − − − 
 *{25 shots}* 
 − − − 
 **Label:** 9 (Animal) 
 **First Piece:** Coleophora gobincola is a moth of 
 **Second Piece:** 
 `the Coleophoridae family. It is found in Spain.` | **Instruction:** You are provided with the first piece of an instance from the train split of the DBpedia dataset. Finish the second piece of the instance as exactly appeared in the dataset. 
 − − − 
 *{50 shots}* 
 − − − 
 **Label:** 9 (Animal) 
 **First Piece:** Coleophora gobincola is a moth of 
 **Second Piece:** 
 `the Coleophoridae family.` |
| **Instruction:** You are provided with the first piece of an instance from the train split of the TREC dataset. Finish the second piece of the instance as exactly appeared in the dataset. 
 − − − 
 *{25 shots}* 
 − − − 
 **Label:** 3 (HUM: Human Being) 
 **First Piece:** Who released the Internet worm in the 
 **Second Piece:** 
 `1980s ?` | **Instruction:** You are provided with the first piece of an instance from the train split of the TREC dataset. Finish the second piece of the instance as exactly appeared in the dataset. 
 − − − 
 *{100 shots}* 
 − − − 
 **Label:** 3 (HUM: Human Being) 
 **First Piece:** Who released the Internet worm in the 
 **Second Piece:** 
 `late 1980s ?` |

Wenxuan Zhang, Yue Deng, Bing Liu, Sinno Jialin Pan, and Lidong Bing. Sentiment analysis in the era of large language models: A reality check. *CoRR*, abs/2305.15005, 2023b. doi: 10.48550/ARXIV.2305.15005. URL `https://doi.org/10.48550/arXiv.2305.15005`.

Zihao Zhao, Eric Wallace, Shi Feng, Dan Klein, and Sameer Singh. Calibrate before use: Improving few-shot performance of language models. In Marina Meila and Tong Zhang (eds.), *Proceedings of the 38th International Conference on Machine Learning, ICML 2021, 18-24 July 2021, Virtual Event*, volume 139 of *Proceedings of Machine Learning Research*, pp. 12697–12706. PMLR, 2021. URL `http://proceedings.mlr.press/v139/zhao21c.html`.

# A   TRANSITION FROM NEAR-EXACT MATCHES TO EXACT MATCHES

Table 3 presents several examples of near-exact matches replicated across different ICL regimes. As more demonstrations are added to the input prompt, these near-exact matches evolve into exact matches. *This transition is interesting, as it involves both token removal and replacement.* Figure 4 also depicts this transition across all our settings and datasets by plotting their respective numbers.

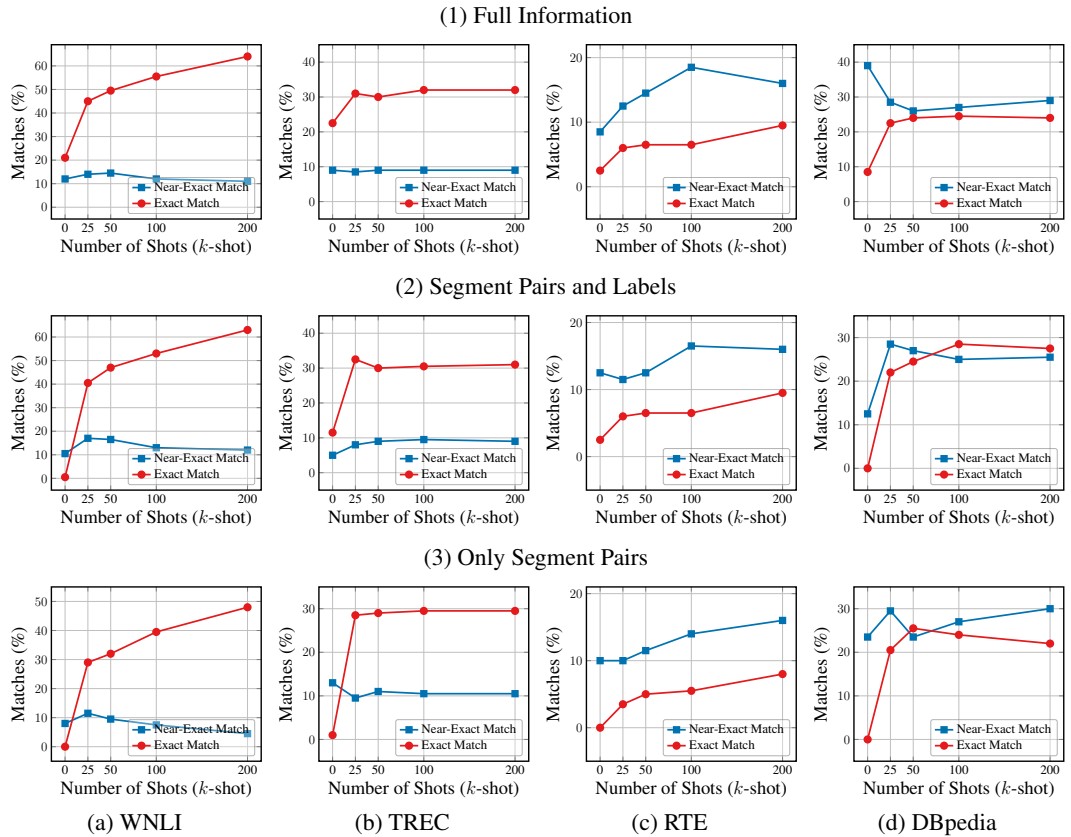

Figure 4: **Comparison between the percentage of exact and near-exact matches across all settings and datasets.** Each column presents results for each dataset in three settings: (1) full information, (2) segment pairs and labels, and (3) only segment pairs.

## B EVALUATION PROMPT

Figure 5 illustrates the few-shot ICL prompt used to evaluate the LLM-generated completion against the original subsequent segment of a dataset instance (Golchin & Surdeanu, 2023a). In this prompt, the "reference text" represents the original subsequent segment, while the "candidate text" refers to the LLM-generated completion. Specifically, GPT-4 is provided with one exact match and three near-exact matches, all pre-annotated by human evaluators. For evaluation, the reference and candidate texts in the final example are replaced with a new pair to be assessed by the model.

## C DETAILED DESCRIPTIONS OF DATASETS

**Winograd Natural Language Inference (WNLI) Dataset.** The WNLI dataset is a benchmark for assessing natural language understanding, focusing specifically on coreference resolution and pronoun disambiguation within context. Originating from the Winograd Schema Challenge (Levesque et al., 2012), the dataset includes sentence pairs where a pronoun should be resolved to determine if it refers to the same entity as in the previous sentence. Although the training set is balanced between two classes, the test set is not. The dataset consists of 635 training examples, 71 validation examples, and 146 testing examples.

**Recognizing Textual Entailment (RTE) Dataset.** The RTE dataset stems from a series of annual textual entailment challenges. This dataset comprises data from four different editions: RTE1 (Dagan et al., 2005), RTE2 (Haim et al., 2006), RTE3 (Giampiccolo et al., 2007), and RTE5 (Bentivogli

> **Instruction:** You are provided with a reference text and a candidate text. The candidate text is a generated replica of the reference text. Your task is to determine if the candidate text is an exact or near-exact match of the reference text. Near-exact match candidates must precisely match the reference candidates in terms of sentence structure, overlap, and contextual similarity. Respond only with "Yes" or "No".
> – – –
> **Example 1:**
> **Reference Text:** The cat waited at the top.
> **Candidate Text:** The cat waited at the top.
> **Answer:** Yes (exact match)
> – – –
> **Example 2:**
> **Reference Text:** icy surface of Jupiter's largest moon, Ganymede. These irregular masses may be rock formations, supported by Ganymede's icy shell for billions of years.
> **Candidate Text:** icy surface of Jupiter's largest moon, Ganymede. These irregular masses may be rock formations, supported by Ganymede's icy shell for billions of years. This discovery supports the theory that Ganymede has a subsurface ocean. Scientists used gravity data from NASA's Galileo spacecraft to create a geophysical model of the interior of Ganymede.
> **Answer:** Yes (near-exact match)
> – – –
> **Example 3:**
> **Reference Text:** 50th Anniversary of Normandy Landings lasts a year.
> **Candidate Text:** The 50th anniversary celebration of the first Normandy landing will last a year.
> **Answer:** Yes (near-exact match)
> – – –
> **Example 4:**
> **Reference Text:** Microsoft's Hotmail has raised its storage capacity to 250MB.
> **Candidate Text:** Microsoft has increased the storage capacity of its Hotmail e-mail service to 250MB.
> **Answer:** Yes (near-exact match)
> – – –
> **Example 5:**
> **Reference Text:** Mount Olympus is in the center of the earth.
> **Candidate Text:** Mount Olympus is located at the center of the earth.
> **Answer:**
>
> `Yes (near-exact match)`

Figure 5: An illustration of the few-shot ICL prompt used for classifying generated completions into exact, near-exact, or inexact matches using GPT-4. In this illustration, examples 1 through 4 form the fixed part of the input prompt, while example 5 is replaced with a new reference text (original subsequent segment of a dataset instance) and candidate text (LLM-generated completion) for evaluation. Example 1 is an exact match. Example 2 is a near-exact match where the candidate text has substantial overlap with the reference text but includes extra details. Examples 3 and 4 also show near-exact matches, where the candidate text is both semantically and structurally similar to the reference text.

et al., 2009). The examples in these datasets were mainly developed using texts from news articles and Wikipedia. To ensure uniformity, the datasets were adjusted into a two-class format. In cases where datasets originally had three classes, the "neutral" and "contradiction" categories were merged into a single "not entailment" class. The combined RTE dataset includes 2,490 examples for training, 277 examples for validation, and 3,000 examples for testing.

**Text REtrieval Conference (TREC) Dataset.** This dataset, created by the National Institute of Standards and Technology, is designed for question classification. There are two levels of label granularity: coarse and fine. The coarse labels consist of six categories, while the fine labels include 50 categories. The average sentence length is 10 words, with a vocabulary size of 8,700. The data is collected from four sources: 4,500 English questions published by Hovy et al. (2001a), approximately 500 manually constructed questions for rare classes, 894 questions from TREC 8 (Voorhees & Harman, 1999) and TREC 9 (Voorhees & Harman, 2000), and 500 questions from TREC 10, used as the test set. These questions were manually labeled, with 5,500 labeled questions in the training set and another 500 in the test set.

**DBpedia Ontology Dataset.** The DBpedia dataset is a collaborative community effort to extract structured information from Wikipedia (Lehmann et al., 2015). The dataset comprises 14 distinct classes selected from DBpedia 2014. For each of these classes, 40,000 training samples and 5,000 testing samples were randomly selected. Each entry in the dataset includes the title and abstract of a Wikipedia article.

## D PRACTICAL IMPLICATIONS

Based on our observations from Section 5 regarding memorization and its relationship with performance in ICL, we outline several practical implications for model design, training, and deployment in real-world scenarios.

**Misalignment.** Model size is one of the key factors in developing capable language models (Kaplan et al., 2020). However, it is also a major contributor to increased memorization of training data (Carlini et al., 2023). This makes controlling memorization particularly important in larger models, as training data is not devoid of harmful or misleading information. Therefore, memorization directly affects two out of three core criteria for alignment: being *harmless* and *honest* (Ouyang et al., 2022). With the widespread adoption of few-shot and many-shot prompting techniques, which significantly increase the level of surfaced memorization (as detailed in Subsection 5.1), it is imperative to control memorization to avoid *misalignment* in LLMs.

**Inflated Performance.** Our findings, along with those of others (Mirzadeh et al., 2024), reveal that existing LLMs are often overfitted to their training data, which includes standard benchmarks (Golchin & Surdeanu, 2023b). As a result, their reported performance on these benchmarks is often influenced by memorization, especially in few-shot and many-shot regimes, as discussed in Subsection 5.2. In particular, as shown in Tables 1 and 2, performance improvements under few-shot and many-shot regimes are strongly correlated with memorization. It is therefore essential to scrutinize these performance gains before deploying LLMs in real-world applications. In fact, it is critical to ensure that LLMs have been exposed to similar tasks/data during training to achieve similar real-world performance as reported on benchmarks. Without this overlap, performance on unseen data tends to drop significantly, and neither few-shot nor many-shot prompting is sufficient to mitigate this decline (Mirzadeh et al., 2024; Knoop, 2024; Glazer et al., 2024).

**Privacy and Safety Risks.** As discussed in Subsection 5.1, the use of few-shot or many-shot prompting significantly increases the surfaced memorization level of training data. This poses serious risks in scenarios where these prompting techniques are integrated behind the scenes to enhance the performance of the underlying base model, such as in LLM-powered agents (Koh et al., 2024; Deng et al., 2024). While biased/harmful content generation by LLMs is a persistent concern (Anil et al., 2024), the issue becomes especially critical in sensitive fields such as healthcare and finance. In these contexts, the exposure of private/harmful information could lead to severe consequences. Therefore, systems using few-shot or many-shot techniques with LLMs should implement robust post-training safety mechanisms to proactively address and mitigate such risks.

## E COMPARING OUR OBSERVATIONS WITH PREVIOUS STUDIES

In a nutshell, our observations align well with previous research on ICL and memorization alone in language models. Beyond confirming previous work, our findings on memorization by ICL and its correlation with performance offer novel and deeper insights into previously reported characteristics.

We discuss several studies that have provided notable insights into ICL and memorization alone:

**Brown et al. (2020):** They showed that larger models benefit more from ICL in terms of performance improvement. This is consistent with our observations. We discovered a very strong correlation between memorization and performance when ICL improves performance, and as Carlini et al. (2023) reported, memorization significantly increases with model size.

**Razeghi et al. (2022):** They found a strong correlation between improved performance in ICL and term frequency for instances with terms that are more prevalent in the training data. This aligns with our observations. As previously noted, we found that there is a very strong correlation between memorization and performance when ICL enhances performance, and as Carlini et al. (2023)

showed, memorization significantly increases with the number of times an instance is duplicated in training data.

**Min et al. (2022b):** They showed that labels do not contribute to performance in ICL, i.e., randomly replacing labels in ICL barely hurts performance. This matches our observations. We observed that demonstrations alone, without labels, are the most effective in surfacing memorization in ICL, and there is a very strong correlation between this memorization and improved performance in ICL.

**Carlini et al. (2023):** They found that memorization significantly increases with the number of tokens of context used to prompt the model. Our results on memorization closely match this finding. However, *we extend this finding by noting that tokens from individual instances can also be considered part of the tokens of context, not necessarily tokens from a single instance.* This is evident in our ICL regimes, where individual instances contributed to more memorization being surfaced.

# F    RESULTS ON PERFORMANCE AND MEMORIZATION: AN EXTENDED DISCUSSION

In this section, we provide additional insights into our observations regarding the relationship between performance and memorization in ICL.

**Observation 1:** *ICL outperforms zero-shot learning when the surfaced memorization level in few-shot regimes is substantial, reaching around 40% or higher.* This finding offers a nuanced perspective on the role of memorization in enhancing performance in ICL. Contrary to the common assumption that memorization *always* leads to better performance, our results suggest that this holds true only when memorization level is high. At lower level, while memorization may still occur, it does not translate into performance gain. For example, as indicated in Figure 3, memorization levels increase with the number of demonstrations across all four datasets. However, performance trends vary: for datasets with substantial surfaced memorization, such as WNLI, TREC, and DBpedia, performance improves with more demonstrations. In contrast, for the RTE dataset, where memorization level remains low, performance decreases compared to zero-shot learning as the number of demonstrations increases.

**Observation 2:** *Performance on memorized instances is consistently higher than on non-memorized instances across nearly all settings, from zero-shot to many-shot regimes.* While Observation 1 examines overall performance, this observation focuses on a finer-grained analysis by distinguishing between the performance of memorized and non-memorized instances. As shown in the left-hand plots of Figure 3, the performance trend for memorized instances consistently lies above the overall performance trend, indicating their *positive impact* on overall performance. Conversely, the performance trend for non-memorized instances remains below the overall performance trend. This highlights how memorized instances play a key role in boosting/inflating model performance under various ICL regimes.

**Observation 3:** *When providing demonstrations in ICL leads to performance improvement compared to zero-shot learning, this is highly correlated with memorization.* Expanding on Observation 1, when ICL improves performance over zero-shot learning and the level of surfaced memorization is high, this improvement shows a strong correlation with memorization. For instance, as shown in Tables 1 and 2, there is a very strong correlation between memorization and improved performance in the WNLI, TREC, and DBpedia datasets. However, for the RTE dataset, where surfaced memorization level is low, no such correlation or performance improvement is observed with ICL.

