# OpenReview forum: "Memorization in In-Context Learning"
_ICLR.cc/2025/Conference — Submitted to ICLR 2025_

### Official Review · Reviewer_bNa5 · 2024-11-02

**Soundness:** 2
**Presentation:** 3
**Contribution:** 2
**Rating:** 6
**Confidence:** 3

**Summary:**

The authors address the memorization problem in the context of In Context Learning.
We recall that memorization is the process of memorizing data from the pre-trained dataset.
It is generally evaluated by measuring the ability of the model to regenerate those data at the inference step.
The main objective of this work is to understand how memorization correlates with Performances in ICL. To this end, the authors propose to study different settings to estimate memorization of the model (using GPT-4 in the experiment), on the dataset used in the pre-training of the model (or that are likely to be seen during its training according to previous works). Authors propose to evaluate different ICL variants (varying between description, example, and labels) in different k-shot settings (number of examples given in the context). For scoring memorization, authors mostly rely on "Time Travel in LLMs: Tracing Data Contamination In Large Language Models," published at ICLR last year.
In the experiments section, authors compare the different memorization scores according to the number k (of k-shots), the information given in input (instruction, example, and/or labels), and the matching method (exact matches, near exact matches).
In subsection 5.2, the correlation between task performance and memorization is compared, showing that the more the dataset is memorized, the more it correlates with performance.
Accordingly, the contributions are the following:
* A new method to measure memorization in ICL, proposing to evaluate memorization with k-shot (the metrics exact match and the near exact match was proposed in [1])
* How to correlate the memorization with performances (in the setting proposed)


[1] "Time Travel in LLMs: Tracing Data Contamination in Large Language Models", Shahriar Golchin and Mihai Surdeanu, ICLR 2024

**Strengths:**

*The originality of the paper proposing a study of memorization impact on performances, while generally it mainly involves detecting contamination
*Experiments are well designed to answer the research question (correlation between performances and memorization)
*A proposal to measure memorization based on a comparison of the number of demonstrations (number of examples in the prompt)

**Weaknesses:**

* Only one model is used in the experiments, which is too few to conclude on a general statement (notice that the choice is, however, justified)
* Most of the pipeline relies on the memorization score design in [1]
* The state-of-the-art lack of explicit explanation (section 6), but contains at the best of my knowledge the most relevant references
* Results should be better/extensively discussed (mainly for the 5.2 experiments)
* Metrics and what the reported results are missing in 5.2 (the two tables 1 and 2)

**Questions:**

* Observation 3 states that performance improvements are highly correlated with memorization. Can you explain why, according to results, particularly how can you explain the negative correlation on RTE
* Tables 1 and 2 report a correlation (Pearson) between performances and memorization; however, the number of k-shots is not specified, nor is the setting (Full Information, Segments pairs, and labels, only segment pairs) to measure the correlation. What did you choose for the memorization metric here?

---

> ### Author Response · Authors · 2024-11-23
> **Official Response by Authors (1/2)**
>
> Thank you for recognizing the originality and effective design of our experimental setup. We are also grateful for your detailed and constructive feedback.
>
> We appreciate the opportunity to address your concerns and questions. To do so, we have provided our responses in the same order in which your concerns and questions were listed.
>
> > Only one model is used in the experiments, which is too few to conclude on a general statement (notice that the choice is, however, justified)
>
> Thank you for your suggestion. We agree that adding more models would enhance our insights. However, we would like to clarify why this option was not feasible for us due to the high cost of our experiments.
>
> First, we did not have sufficient GPU resources to conduct experiments with long-content input prompts. As a result, we focused on proprietary LLMs where GPU power is supported by the model provider. As mentioned in Section 4 under Model, we conducted a pilot study with existing proprietary LLMs and selected GPTs based on the results. Initially, we aimed to control costs by using GPT-3.5. However, its 16k context limit prevented us from conducting experiments in many-shot regimes.
>
> To ensure comprehensive experiments across various regimes (i.e., zero-shot, few-shot, and many-shot), we managed our budget to use GPT-4, which offers a 32k context length, enabling many-shot experimentation. However, these experiments were quite expensive, making it infeasible to test additional closed- or open-weight LLMs. To give you an idea of the costs involved, OpenAI charges for API usage based on both **input** and **output** tokens. While the output tokens in our experiments were negligible, the input tokens were the primary cost driver. For example, in a 200-shot scenario, the input tokens ranged from 15k to 22k. OpenAI charges 60 dollars per 1 million input tokens, meaning each API call costs approximately 1 dollar in a 200-shot scenario. _For every data point in the left-hand plots of Figure 2, we tested 200 instances, resulting in a cost of 200 dollars per data point for 200-shot experiments._ Considering that *we repeated these experiments across four datasets and three different settings (full information, segment pairs and labels, and only segments),* the experiments became very expensive. In fact, such an undertaking is very difficult without industry-level resources.
>
> Finally, we hope this explanation justifies the budget constraints that led to our use of GPT-4 in our experiments.
>
> > Most of the pipeline relies on the memorization score design in [1]
>
> Thank you for your input. We would like to argue that our reliance on and adaptation of the methodology proposed in [1] offers several advantages, making it well-suited for our use case. First, the intuitive nature of the method ensures ease of use, thereby promoting broader adoption within the research community and facilitating further exploration. Second, the method's strategy for formatting input prompts for quantifying memorization levels in a k-shot setting is identical to standard ICL, where k demonstrations are added to the input prompt. This makes it simple to compute memorization levels across different scenarios, ranging from zero-shot to many-shot. Third, the method is both task- and model-agnostic, making it applicable to any task or dataset for memorization analysis. Therefore, we believe that while the method in [1] is highly effective and adaptable, it also provides features that make it universally applicable and generalizable across all scenarios, which is essential for its widespread use within the community.
>
> > The state-of-the-art lack of explicit explanation (section 6), but contains at the best of my knowledge the most relevant references
>
> Thank you for your feedback and for recognizing our effort to include the most relevant literature. We would like to note that we tried our best to incorporate as many details as possible within the space constraints. We also strived to include the most relevant and recent literature in our Related Work section, including papers published this year. Given the dynamic nature of the field and the continuous publication of new papers, it is inevitable that some studies may have been missed. If you have any specific papers in mind that are relevant to our research, please let us know, and we will surely consider including them in the Related Work section.
>
> > Results should be better/extensively discussed (mainly for the 5.2 experiments)
>
> Thank you for your suggestion. __We have added a new section, Appendix F, to the current revised version of the paper to provide further details on our findings from Subsection 5.2.__ The newly added text is highlighted in blue.

---

> > ### Author Response · Authors · 2024-11-23
> > **Official Response by Authors (2/2)**
> >
> > > Metrics and what the reported results are missing in 5.2 (the two tables 1 and 2)
> >
> > Thank you for your feedback. We would greatly appreciate it if you could clarify which specific metrics are missing in Subsection 5.2. Upon our review, we could not identify any missing metrics in Subsection 5.2, and Tables 1 and 2. In particular, we have discussed Figure 3 and Tables 1 and 2 extensively within Subsection 5.2, ensuring that each includes detailed titles and clearly labeled y-axes and column names to reflect the metrics of the reported results. Your further guidance would be immensely helpful.
> >
> > __Questions:__
> >
> > > Observation 3 states that performance improvements are highly correlated with memorization. Can you explain why, according to results, particularly how can you explain the negative correlation on RTE
> >
> > Thank you for your question. As you mentioned above, and as highlighted in our Observation 3, __when ICL results in performance improvement, it is highly correlated with memorization.__ _However, in the RTE dataset, ICL does not lead to any performance improvement compared to zero-shot learning._ Specifically, all ICL regimes, whether few-shot or many-shot, underperform zero-shot learning. In this case, **since ICL does not enhance performance, the correlation with memorization is negative.**
> >
> > > Tables 1 and 2 report a correlation (Pearson) between performances and memorization; however, the number of k-shots is not specified, nor is the setting (Full Information, Segments pairs, and labels, only segment pairs) to measure the correlation. What did you choose for the memorization metric here?
> >
> > Thank you for your question. As detailed in Subsection 3.4 and illustrated in Figures 2 and 3, the k-shot scenarios used were k = {0, 25, 50, 100, 200}. Additionally, both Tables 1 and 2 specify the setting in the first __column entitled "Setting"__ and include detailed titles describing the metric used for quantifying memorization. Specifically, Table 1 lists Pearson correlation coefficients when memorization is quantified using both exact and near-exact matches, while Table 2 reports correlations when memorization is quantified using only exact matches.
> >
> > ---
> >
> > Finally, we hope our responses have fully addressed your concerns and questions. We have tried our best to be as thorough as possible. If you have any further questions or feel that something needs more explanation, please let us know. We would be more than happy to discuss anything in more detail.

---

> ### Author Response · Authors · 2024-12-01
> **Kind Reminder About Our Rebuttals**
>
> Dear Reviewer bNa5,
>
> Thank you very much for taking the time to review our paper. As the rebuttal period is coming to an end, _we would like to kindly remind you that we have carefully and thoroughly responded to all your questions and concerns, and added a new section (Appendix F) to the paper to further elaborate on our findings from Subsection 5.2._ We deeply value your feedback and would greatly appreciate it if you could provide any additional input or confirm whether our responses have addressed your concerns.
>
> Thank you for your time and attention.
>
> Best regards,
> The Authors

---

### Official Review · Reviewer_Czjk · 2024-11-04

**Soundness:** 2
**Presentation:** 3
**Contribution:** 2
**Rating:** 5
**Confidence:** 3

**Summary:**

The paper explores the phenomenon of memorization in In-context Learning (ICL) across different regimes (zero-shot, few-shot, and many-shot) using large language models (LLMs). It aims to quantify how much of the model's performance improvement during ICL can be attributed to memorized training data.

**Strengths:**

Relevant Topic: The topic of memorization in LLMs is timely and relevant, given the increasing reliance on these models for various tasks without full retraining.

Experimental Design: The approach to quantify memorization using modified data contamination methods is methodologically interesting and innovative.

**Weaknesses:**

Incremental Contribution: The insights and contributions of the paper are not sufficiently novel or significant. The findings that ICL surfaces memorization and that memorization correlates with performance do not extend significantly beyond what is already suggested or known in the literature.

Lack of Practical Implications: The paper does not sufficiently discuss the practical implications of its findings. While it notes the correlation between memorization and performance, it fails to explore how these insights could be used to improve model design or deployment in real-world applications.

Theoretical Depth: The discussion around why certain memorization occurs in specific ICL settings lacks depth. There is no substantial theoretical analysis or explanation beyond the observational data presented.

**Questions:**

Can the authors provide more detailed insights into how these findings could influence practical model training or deployment strategies? How do the authors envision their methodology being adapted or expanded to cover more diverse models and tasks to verify the generalizability of the findings?

---

> ### Author Response · Authors · 2024-11-23
> **Official Response by Authors (1/2)**
>
> Thank you for highlighting the significance of our work and for finding our experimental design innovative and interesting. We also appreciate your thoughtful and constructive review.
>
> We appreciate the chance to address your concerns and respond to your questions. Below, please find our responses presented in the same order in which your concerns and questions were mentioned.
>
> > Incremental Contribution: The insights and contributions of the paper are not sufficiently novel or significant. The findings that ICL surfaces memorization and that memorization correlates with performance do not extend significantly beyond what is already suggested or known in the literature.
>
> Thank you for your feedback. We would like to clarify the novelty and significance of our work from different perspectives.
>
> __First, our study is the first to identify memorization as a factor impacting ICL.__ Prior to our research, no studies had indicated that ICL could potentially increase memorization behind the scenes. __Second, we proposed a novel methodology to detect and quantify this memorization, which can be implemented and scaled as easily as standard ICL.__ __Third, we are the first to measure the correlation between surfaced memorization in ICL and improved performance.__ To the best of our knowledge, no previous work has examined memorization in ICL. We would appreciate any references to previous literature with similar findings on surfaced memorization in ICL, as this would help us better compare our work and highlight its contributions and significance.
>
> Furthermore, **we would like to emphasize that the surfaced memorization by ICL does not always correlate with performance, as demonstrated by our results from the RTE dataset in Figure 3 and Tables 1 and 2.** In fact, this is one of our key findings that when memorization does not reach a substantial level, ICL fails to outperform zero-shot learning. With this in mind, we strongly believe that no prior work has presented such findings.
>
> > Lack of Practical Implications: The paper does not sufficiently discuss the practical implications of its findings. While it notes the correlation between memorization and performance, it fails to explore how these insights could be used to improve model design or deployment in real-world applications.
>
> Thank you for your suggestion. __We have added a new section, Appendix D, to the current revised version of the paper to discuss the practical implications of our findings.__ The newly added text is highlighted in blue.
>
> > Theoretical Depth: The discussion around why certain memorization occurs in specific ICL settings lacks depth. There is no substantial theoretical analysis or explanation beyond the observational data presented.
>
> Thanks for the suggestion. We would like to emphasize that the challenge of explaining the underlying reasons for observed experimental results is a widely acknowledged limitation in the field of Deep Learning, especially concerning large and deep models. _In fact, this is not specific to our work; it extends to many studies in this area, given the limited theoretical foundation currently available._ As such, much of the research, including ours, is necessarily practical in nature.
>
> For example, in the relevant and highly regarded work of Carlini et al. (2023) on quantifying memorization in language models, no definitive reasons are provided regarding why memorization occurs or why there is a log-leaner relationship between memorization and other factors, such as model size. Instead, such studies present practical observations aimed at shedding light on the behavior of models, without delving into causal explanations.
>
> Moreover, the inherent complexity of current models, the opaque nature of deep learning processes, and limited transparency concerning the data and training paradigms add further layers of difficulty to understanding the underlying reasons behind these phenomena.
>
> _In our case, providing reasons for the observed findings would necessitate a precise understanding of why memorization occurs in language models and the exact mechanisms governing ICL. To the best of our knowledge, no existing works have conclusively established such explanations._ **Therefore, we are constrained by the current state of knowledge in providing definitive underlying reasons for our observed findings.**
>
> __Questions:__
>
> > 1. Can the authors provide more detailed insights into how these findings could influence practical model training or deployment strategies?
>
> Thank you for your question. We answered this before. Please refer to our response above and in __Appendix D__ of the current revised version of the paper.

---

> > ### Author Response · Authors · 2024-11-23
> > **Official Response by Authors (2/2)**
> >
> > > 2. How do the authors envision their methodology being adapted or expanded to cover more diverse models and tasks to verify the generalizability of the findings?
> >
> > Thank you for your question. We would like to state that our findings and methodology can be generalized to any models and tasks. This generalizability stems from our method's use of demonstrations as straightforward as standard ICL in order to quantify memorization. This inherent flexibility means our approach is not limited to specific models or data types, similar to how standard ICL can be utilized across nearly all tasks.
> >
> > Additionally, the criteria we established in the paper for studying memorization (Subsections 3.5 and 3.6) in ICL were designed to systematically address our research questions. However, this does not restrict the generalizability of our findings and methodology to other models and tasks, as memorization is present in all language models (Carlini et al., 2023), and as mentioned earlier, ICL can be used across nearly all tasks, supporting the generalizability of our work.
> >
> > ---
> >
> > In closing, we hope our responses have addressed your concerns effectively. We are always happy and open to further discussion if you still have any concerns or questions.
> >
> > ---
> >
> > References:
> >
> > [1] Quantifying Memorization Across Neural Language Models (Carlini et al., ICLR 2023)

---

> ### Author Response · Authors · 2024-12-01
> **Kind Reminder About Our Rebuttals**
>
> Dear Reviewer Czjk,
>
> Thank you very much for taking the time to review our paper. As the rebuttal period is coming to an end, _we would like to kindly remind you that we have carefully and thoroughly responded to your concerns and added a new section (Appendix D) to the paper to discuss the practical implications of our findings._ We deeply value your feedback and would greatly appreciate it if you could provide any additional input or confirm whether our responses have addressed your concerns.
>
> Thank you for your time and attention.
>
> Best regards,
> The Authors

---

### Official Review · Reviewer_jwx1 · 2024-11-05

**Soundness:** 3
**Presentation:** 4
**Contribution:** 4
**Rating:** 8
**Confidence:** 4

**Summary:**

The paper seeks to quantify the correlation between LLM memorization of specific datapoints in well-known datasets and the ICL performance on these dataset subsets. They examine several datasets, selecting datapoints from these and using an existing protocol from another work to quantify how many datapoints the LLM has memorized, testing memorization in few, many, and zero-shot regimes. They then correlate this to the performance using ICL on these exact datapoints, but overall in aggregate, rather than on a per-datapoint level. Authors then draw a set of observations from this experiment and conclude that the results raise the question of if ICL works by memorization.

**Strengths:**

The paper is very well written. It is clear and easily to follow, and the experimental setup is also very intuitive. The mechanism by which ICL works is a very relevant and important question.

**Weaknesses:**

The paper only experiments on GPT-4. Authors claim that it is the only LLM that fulfills their criteria but this is somewhat hard to believe, especially given the existence of long-context open-source models. The authors claim that they do not have the resources to run these experiments on e.g. llama3 or some other long-context open-source model that fulfills their criteria, but I believe it would strengthen the paper considerably to have more than a single model for testing.

It is not clear what authors provide beyond confirming existing works in the area. It is not justified adequately why these experiments they perform give us any new information about how ICL fuctions.

**Questions:**

Use additional models, it's hard to take a study that only uses a single closed-source model seriously.

I need additional justification for why these experiments tell us about how ICL functions. To me they tell me that in cases where there is dataset contamination ICL performs better, but this is to be expected. What about when there is no dataset contamination and ICL still works, how does this work? This is the more interesting question to me.

---

> ### Author Response · Authors · 2024-11-23
> **Official Response by Authors (1/2)**
>
> Thank you for highlighting the clarity of our presentation, as well as the straightforwardness and importance of our work. We also appreciate your precise and insightful feedback.
>
> We are grateful for the opportunity to address your concerns and questions. Below, we provide our responses to your comments and questions in the order they were presented.
>
> > The paper only experiments on GPT-4. Authors claim that it is the only LLM that fulfills their criteria but this is somewhat hard to believe, especially given the existence of long-context open-source models. The authors claim that they do not have the resources to run these experiments on e.g. llama3 or some other long-context open-source model that fulfills their criteria, but I believe it would strengthen the paper considerably to have more than a single model for testing.
>
> Thank you for your suggestion. We agree that adding more models would enhance our insights. However, we would like to clarify why this option was not feasible for us due to the high cost of our experiments.
>
> First, we did not have sufficient GPU resources to conduct experiments with long-content input prompts. As a result, we focused on proprietary LLMs where GPU power is supported by the model provider. As mentioned in Section 4 under Model, we conducted a pilot study with existing proprietary LLMs and selected GPTs based on the results. Initially, we aimed to control costs by using GPT-3.5. However, its 16k context limit prevented us from conducting experiments in many-shot regimes.
>
> To ensure comprehensive experiments across various regimes (i.e., zero-shot, few-shot, and many-shot), we managed our budget to use GPT-4, which offers a 32k context length, enabling many-shot experimentation. However, these experiments were quite expensive, making it infeasible to test additional closed- or open-weight LLMs. To give you an idea of the costs involved, OpenAI charges for API usage based on both **input** and **output** tokens. While the output tokens in our experiments were negligible, the input tokens were the primary cost driver. For example, in a 200-shot scenario, the input tokens ranged from 15k to 22k. OpenAI charges 60 dollars per 1 million input tokens, meaning each API call costs approximately 1 dollar in a 200-shot scenario. _For every data point in the left-hand plots of Figure 2, we tested 200 instances, resulting in a cost of 200 dollars per data point for 200-shot experiments._ Considering that *we repeated these experiments across four datasets and three different settings (full information, segment pairs and labels, and only segments),* the experiments became very expensive. In fact, such an undertaking is very difficult without industry-level resources.
>
> Finally, we hope this explanation justifies the budget constraints that led to our use of GPT-4 in our experiments.
>
> > It is not clear what authors provide beyond confirming existing works in the area. It is not justified adequately why these experiments they perform give us any new information about how ICL fuctions.
>
> Thank you for your input. We address this in detail in the following, specifically in response to the second question, where we explain how our experiments offer greater insight into how ICL functions.
>
> __Questions:__
> > Use additional models, it's hard to take a study that only uses a single closed-source model seriously.
>
> We answered this before. Please find our response above.

---

> ### Author Response · Authors · 2024-11-23
> **Official Response by Authors (2/2)**
>
> > I need additional justification for why these experiments tell us about how ICL functions. To me they tell me that in cases where there is dataset contamination ICL performs better, but this is to be expected. What about when there is no dataset contamination and ICL still works, how does this work? This is the more interesting question to me.
>
> Thank you for your question. We would like to clarify that our decision to experiment with datasets that were part of the training data aligns with our research objectives, as outlined in the paper (lines 36-38). From a technical perspective, studying the memorization of data that was not included in the training process would not be meaningful. **Moreover, it is important to highlight that ICL *does not always perform better* for datasets included in the training data. Our experiments with the RTE dataset, shown in Figure 2, demonstrate that adding demonstrations did not improve performance compared to zero-shot learning.** In contrast, ICL tends to enhance performance in cases where memorization levels are high, such as in the scenarios observed with WNLI, TREC, and DBpedia datasets.
>
> Regarding your question about scenarios where datasets are not part of the training data but ICL remains effective, we would like to direct you to the work of Mirzadeh et al. (2024). In brief, based on our findings on memorization in ICL and the insights provided by this study, it appears that in such cases, models attempt to retrieve similar/relevant information from their memorized training data. However, we were not able to use such a setting in our experiments, as quantifying memorization for data that was not part of the model’s training data would be meaningless.
>
> ---
>
> Lastly, we hope our explanations have addressed your concerns effectively. While we have aimed to cover every aspect thoroughly, we are happy to provide further clarification or expand on any topics if you still have questions or feel our responses were not convincing.
>
> ---
>
> References:
>
> [1] GSM-Symbolic: Understanding the Limitations of Mathematical Reasoning in Large Language Models (Mirzadeh et al., arXiv 2024)

---

> ### Comment · Reviewer_jwx1 · 2024-11-24
>
> I appreciate your responses. I accept the analysis of cost as a justification for using GPT-4 only. However, I am not convinced w.r.t. my other concerns, esp. re: novelty compared to existing works. The main work that heavily implies ICL is connected to memorization, reducing the novelty of your claim is [1]. As other authors mentioned, the fact that LLMs memorize training data is also heavily implied by several other works ([2], [3]), and even surveys on the subject ([4]).
>
> [1] https://arxiv.org/abs/2202.07206
>
> [2] https://arxiv.org/abs/2211.08411
>
> [3] https://arxiv.org/abs/2012.07805
>
> [4] https://arxiv.org/abs/2410.02650
>
> The novel claim you are making is that you are the first to investigate the relationship between *in-context learning* and memorization. The conclusion that "ICL tends to enhance performance in cases where memorization levels are high" (implied to be related to frequency of seeing the training data of the dataset in the pre-training process by all the above works), again seems to be self-evident to me. Can you explain why the insight "when LLMs memorize training instances from their pre-training process, they also tend to memorize the accompanying labels" is not self-evident?

---

> > ### Author Response · Authors · 2024-11-25
> > **Official Response by Authors**
> >
> > Thank you for your question and engagement during the rebuttal period. We are happy to clarify the novelty of our work in comparison to [1]. *We believe the misunderstanding arises from the way the comparison is framed.*
> >
> > Regarding our conclusion that "ICL tends to enhance performance in cases where memorization levels are high" and implying that this can be related to the frequency of seeing the training data of the dataset in the pre-training process (or term frequency, for short), **we need to clarify that the observed performance improvements in [1] are not tied to an isolated model.** In other words, when analyzing memorization through the lens of term frequency for **a standalone model,** we need to consider a constant term frequency, which means a single point on the x-axis in the plots presented in [1]. *At that point, we can then examine whether term frequency can explain performance improvements through memorization or not.* However, in this scenario, it does not, because term frequency remains constant. On the other hand, our work highlights that with a constant term frequency in a standalone model, the level of memorization varies based on the number of demonstrations provided in ICL. This observation cannot be explained by any previous literature.
> >
> > Regarding your question, the reason the insight that "when LLMs memorize training instances from their pre-training process, they also tend to memorize the accompanying labels" is __not self-evident__ is that many dataset instances are sourced from internet data and are labeled *afterward.* Therefore, memorizing training instances does not necessarily imply that the accompanying labels are also memorized, as the training instances alone can exist in internet data and can be memorized by the model without any associated labels.
> >
> > Thank you again for your engagement. Please let us know if you have any further questions or need additional clarification. We would be happy to discuss and provide more details.

---

> > > ### Comment · Reviewer_jwx1 · 2024-11-25
> > >
> > > Thank you for your reply, I appreciate the engagement.
> > >
> > > By providing more examples via ICL, it intuitively makes sense to me that you would "surface" more evidence of memorization (a longer context would allow the LM to "pinpoint" in a sense more effectively what training data we're referring to in the context). If I'm not mistaken, this is indeed what you observe (in all except the one RTE case). However, fundamentally we're still talking about the relationship between dataset memorization and performance on said dataset. What does this tell us fundamentally about how ICL works? I'm still struggling see the contribution.
> > >
> > > So if I understand correctly, the argument is that the training data for these datasets could have appeared in the pre-training data independently of the labels (in the sense that the datasets are potentially formed from excerpts in the training data, i.e. scraped from the internet), in which case the degree of memorization *of the training instances* does not necessarily correspond to the memorization of the labels. But how many of the datasets you tested were actually formed in this way, and you can confirm that the instances appeared separate from the labels? In this case using an open-source model with open training data (something like AllenAI's OLMo), would allow you to perform this kind of analysis.
> > >
> > > Furthermore, can authors explain a bit more the intention behind the comparison with zero-shot learning?

---

> > > > ### Author Response · Authors · 2024-11-25
> > > > **Official Response by Authors**
> > > >
> > > > Thank you very much for your reply. Please find our responses below.
> > > >
> > > > > By providing more examples via ICL, it intuitively makes sense to me that you would "surface" more evidence of memorization (a longer context would allow the LM to "pinpoint" in a sense more effectively what training data we're referring to in the context). If I'm not mistaken, this is indeed what you observe (in all except the one RTE case). However, fundamentally we're still talking about the relationship between dataset memorization and performance on said dataset. What does this tell us fundamentally about how ICL works? I'm still struggling see the contribution.
> > > >
> > > > We would like to express some skepticism regarding the intuitiveness of the idea that providing additional examples surfaces more evidence of memorization. This is because *each example is independent,* and no previous work has demonstrated how these independent examples collectively lead to a higher level of memorization. In fact, based on the findings of Carlini et al. (2023), such an increase in memorization should not occur in this case, *as examples in ICL are independent.* The scenario you referred to applies only when the context is derived from a __single instance, not from multiple independent instances.__
> > > >
> > > > Regarding our observations about the increase in memorization levels, we would like to emphasize that this trend is consistent across all datasets, __including the RTE dataset.__ This is shown in the plots in Figure 2 of our paper. Additionally, we respectfully disagree with the characterization of our research as primarily focusing on the relationship between dataset memorization and performance. __While this relationship _may_ be relevant in our zero-shot experiments, it does not extend to our few-shot or many-shot experiments.__
> > > >
> > > > Moreover, while it is impossible to make definitive claims about how ICL works given the current state of research, _our findings suggest that memorization plays a significant behind-the-scenes role in ICL._ __Also, in terms of contribution, to the best of our knowledge, no prior work has (1) detected memorization in ICL, (2) quantified this memorization, or (3) computed its correlation with performance.__ These are the key contributions we present in our paper, addressing areas that were previously unexplored.
> > > >
> > > > > So if I understand correctly, the argument is that the training data for these datasets could have appeared in the pre-training data independently of the labels (in the sense that the datasets are potentially formed from excerpts in the training data, i.e. scraped from the internet), in which case the degree of memorization of the training instances does not necessarily correspond to the memorization of the labels. But how many of the datasets you tested were actually formed in this way, and you can confirm that the instances appeared separate from the labels? In this case using an open-source model with open training data (something like AllenAI's OLMo), would allow you to perform this kind of analysis.
> > > >
> > > > Thank you for your suggestion about using open-source models to confirm whether dataset instances appeared in the pre-training data without labels. While we appreciate the idea, we must emphasize that this is not an easy task, even with industry-level resources and current methods.
> > > >
> > > > At first glance, the process might seem straightforward, but cross-verifying each data point with pre-training data requires immense computational power, even when using approximate methods like high-order n-grams (e.g., 13-grams). Even if the necessary resources are available, these methods are inherently approximate. As a result, the outcome would only provide an estimated percentage of such data points, rather than a definitive one.
> > > >
> > > > To illustrate this complexity, we refer to the work of Brown et al. (2020) on training GPT-3. In Subsection 2.2 of their paper, the authors explicitly state that they could not successfully filter their training data. This highlights the challenges involved. Given the vast scale of pre-training data and the computational demands, conducting such experiments is far from straightforward. Even with access to pre-training data and the availability of the required computational resources, the results would remain estimates rather than exact figures.
> > > >
> > > > > Furthermore, can authors explain a bit more the intention behind the comparison with zero-shot learning?
> > > >
> > > > Thank you for the question. We would like to emphasize that without comparing our ICL experiments to zero-shot learning, it was unclear to what extent ICL contributes to both memorization and performance. In fact, ICL is consistently used as an enhanced method compared to zero-shot learning. Thus, to assess this contribution in terms of both memorization and performance, we included zero-shot learning in our experiments.
> > > >
> > > > Please let us know if you have any further questions or need additional clarification. We would be happy to discuss and provide more details.

---

> ### Comment · Reviewer_jwx1 · 2024-11-25
>
> Hello, thanks again for your responses.
>
> I'm not sure I'm sold on the idea that each dataset instance is independent, when it comes to the pre-training process. Is it not relatively likely that the dataset instances appeared as a set all together in the pre-training text and therefore the model implicitly learned them as a single context? It is not particularly clear to me from Carlini et al. (2023) that this "independent datapoints" idea actually holds. In fact, this is where I think studying a model like AllenAI's OLMo with open pre-training data would be particularly helpful.
>
> The points about this analysis not being an easy task is a fair point to make. However projects like OLMo are specifically meant to address this difficulty, providing specialized tooling meant to facilitate searching through the pre-training data [1]. I unfortunately still have to echo the sentiments of the other reviewers in saying that with the current experimental setup in the paper, using a single closed-source unknown-pretraining-data model, it's very hard to draw generalized conclusions from this work, and therefore it still feels like it is of somewhat limited utility.
>
> Specifically, let's look at each of the claims in the abstract under the lens of what I mentioned in the first paragraph, that it is reasonable to expect that dataset instances appeared together in the training data.
>
> Claim 1: ICL significantly surfaces memorization compared to zero-shot learning in most cases
>
> - this is a reasonable logical deduction from Carlini et al. (2023)'s "more context surfaces more memorization", as in the ICL context you are giving significantly more tokens to the model than in the zero-shot case
>
> Claim 2: demonstrations, without their labels, are the most effective element in surfacing memorization
>
> - I'm not sure why this is an interesting claim to be honest. It logically follows if more context surfaces more memorization that the model would be able to surface the memorized instances and just not include the labels in its output, if it's following an existing pattern of omitting the labels in the previous context it's seen
>
> Claim 3: ICL improves performance when the surfaced memorization in few-shot regimes reaches a high level
> (about 40%)
>
> - this is the main interesting claim to me, effectively implying that ICL is driven primarily by memorization. However, as mentioned before, there are still significant concerns
>
> Claim 4: there is a very strong correlation between performance and memorization in ICL when it outperforms zero-shot learning
>
> - no comment here, although the distinction between this and Claim 3 is hazy to me.
>
> [1] https://github.com/allenai/dolma

---

> ### Author Response · Authors · 2024-11-28
> **Official Response by Authors (1/2)**
>
> Thank you very much again for your participation and engagement during the rebuttal period. Please find our responses in the following.
>
> > I'm not sure I'm sold on the idea that each dataset instance is independent, when it comes to the pre-training process. Is it not relatively likely that the dataset instances appeared as a set all together in the pre-training text and therefore the model implicitly learned them as a single context? It is not particularly clear to me from Carlini et al. (2023) that this "independent datapoints" idea actually holds. In fact, this is where I think studying a model like AllenAI's OLMo with open pre-training data would be particularly helpful.
>
> Thank you for your input. To clarify the concept of the independence of each dataset instance, we refer you to Definition 3.1 from the work of Carlini et al. (2023). According to this definition, a string $s$ is considered extractable if the model $f$ can generate it when prompted with string $p$ (prefix), where the concatenation $[p||s]$ exists in the training data.
>
> This definition clearly shows that the experiments conducted by Carlini et al. (2023) were based on independent data points. __Otherwise, $p$ could serve as a prefix, suffix, or any part of $s$, and not exactly a prefix.__ However, in our ICL experiments, we used a single prompt containing a fixed set of random demonstrations for each $k$-shot setting, where each of these demonstrations appeared in the pertaining data in any arbitrary order relative to the downstream instance being replicated during inference. _This distinction highlights the difference between our findings and those reported by Carlini et al. (2023)._ __Our results demonstrate that $p$ can be a prefix, suffix, or any part of the context, contributing to memorization.__
>
> > The points about this analysis not being an easy task is a fair point to make. However projects like OLMo are specifically meant to address this difficulty, providing specialized tooling meant to facilitate searching through the pre-training data [1]. I unfortunately still have to echo the sentiments of the other reviewers in saying that with the current experimental setup in the paper, using a single closed-source unknown-pretraining-data model, it's very hard to draw generalized conclusions from this work, and therefore it still feels like it is of somewhat limited utility.
>
> Thank you for your feedback. However, it is unclear to us which part of our conclusions requires access to the pertaining data. Furthermore, we would like to emphasize that most research in the field is currently conducted using closed-data LLMs while their weights are accessible (e.g., LLaMA, Mistral, etc). __Therefore, we believe it is unfair to critique our work on the basis that we used a closed-data model. By this reasoning, most papers currently being submitted or previously published would lack generalizability, as they also rely on closed-data models.__
>
> _As a result, we respectfully do not find this review and judgment to be valid/fair._
>
> > Claim 1: ICL significantly surfaces memorization compared to zero-shot learning in most cases
> > - this is a reasonable logical deduction from Carlini et al. (2023)'s "more context surfaces more memorization", as in the ICL context you are giving significantly more tokens to the model than in the zero-shot case
>
> Thank you for your input. While we have addressed this before, we will reiterate our response for clarity. In the work of Carlini et al. (2023), based on Definition 3.1, the context is defined by $p$, where $p$ represents a _prefix_ of a sequence. In our work, however, this $p$ can be a prefix, a suffix, or any part of the context. We hope this explanation clearly highlights the difference between the findings provided by Carlini et al. (2023) and the findings in our work.
>
> > Claim 2: demonstrations, without their labels, are the most effective element in surfacing memorization
> > - I'm not sure why this is an interesting claim to be honest. It logically follows if more context surfaces more memorization that the model would be able to surface the memorized instances and just not include the labels in its output, if it's following an existing pattern of omitting the labels in the previous context it's seen
>
> Thank you for your feedback. This is an interesting observation, as it helps explain previous findings related to ICL. For instance, Min et al. (2022) found that labels have little effect on performance in ICL. Similarly, Agarwal et al. (2024) showed that in question-solution tasks, ICL enhances performance even when the model is prompted only with questions, _without the corresponding solutions._ __Our work is especially significant because it sheds light on the underlying reasons for these findings by examining the role of memorization.__

---

> > ### Author Response · Authors · 2024-11-28
> > **Official Response by Authors (2/2)**
> >
> > > Claim 3: ICL improves performance when the surfaced memorization in few-shot regimes reaches a high level (about 40%)
> > > - this is the main interesting claim to me, effectively implying that ICL is driven primarily by memorization. However, as mentioned before, there are still significant concerns
> >
> > Thank you for expressing your interest in this observation. This observation was based on _four datasets, with three of them exhibiting this behavior,_ and this behavior was consistent among all where ICL enhanced their performance. Please let us know if you have any further concerns.
> >
> > ---
> > References:
> >
> > [1] Rethinking the Role of Demonstrations: What Makes In-Context Learning Work? (Min et al., EMNLP 2022)
> >
> > [2] Many-Shot In-Context Learning (Agarwal et al., NeurIPS 2024)

---

> ### Comment · Reviewer_jwx1 · 2024-11-29
>
> >This distinction highlights the difference between our findings and those reported by Carlini et al. (2023).
>
> I thank the authors for the explicit clarification -- this makes sense.
>
> >However, it is unclear to us which part of our conclusions requires access to the pertaining data.
>
> Access to the pre-training data would allow you to quantify the degree of "contamination", which seems like it could be a useful factor for analysis.
>
> To recognize the authors' consistent engagement with the Discussion process and addressing the majority of my concerns, and after looking at the latest version of the PDF, I have raised my rating of the paper from a 6 to an 8, the soundness from a 2 to a 3, and the contribution from a 3 to a 4. I commend the authors for their great work in improving the paper, especially the additional discussion in the Appendices.

---

> > ### Author Response · Authors · 2024-11-29
> > **Official Response by Authors**
> >
> > We are glad that our responses have addressed your concerns. We also deeply appreciate your valuable and constructive feedback, which has significantly enhanced our work. Lastly, we sincerely thank you for taking the time to actively and consistently participate and engage during the rebuttal period.

---

### Official Review · Reviewer_QHTo · 2024-11-07

**Soundness:** 3
**Presentation:** 3
**Contribution:** 2
**Rating:** 5
**Confidence:** 3

**Summary:**

This paper examines the role of memorization in in-context learning (ICL) for large language models (LLMs), investigating how ICL surfaces memorized training data and analyzing its impact on downstream task performance across zero-shot, few-shot, and many-shot regimes. The authors propose a method to measure memorization by testing if model-generated completions match known dataset instances. They find that ICL significantly increases memorization compared to zero-shot, with the amount of memorized information correlating strongly with performance improvement. This study highlights memorization as a key factor impacting ICL effectiveness, prompting questions about the extent to which LLMs generalize from demonstrations versus relying on memorized knowledge.

**Strengths:**

1.	The paper is the first to systematically examine the relationship between ICL and memorization in LLMs, providing new insights into how memorized knowledge influences ICL performance.
2.	The study uses a detailed approach to measure memorization across multiple settings (full information, segment pairs and labels, and only segment pairs), allowing for a granular analysis of which prompt elements drive memorization in ICL.
3.	The paper demonstrates a robust correlation between memorization and improved performance in downstream tasks, highlighting memorization as a previously under-explored factor in ICL success.
4.	The methodology and results are clearly presented, making the findings accessible and useful for future research on optimizing ICL methods and evaluating LLM generalization.

**Weaknesses:**

1.	While the experiments are thorough, they are conducted in relatively simple datasets, limiting the paper’s ability to generalize findings to more complex, real-world tasks (e.g., legal, medical datasets).
2.	The study does not address potential challenges in handling longer contexts, which are often needed in real-world applications and may limit the practicality of the proposed memorization detection method in large-scale LLMs.
3.	While the paper successfully identifies memorization as a factor, it does not provide a concrete analysis of how much ICL performance improvement is due to memorization versus actual generalization, leaving this as an open question.
4.	The experiments rely solely on GPT-4, limiting the generalizability of the findings to other LLMs. The authors could strengthen their conclusions by evaluating memorization across a range of models with varying training data and architectures.

**Questions:**

1.	How would the memorization-performance correlation change if tested on larger, more diverse datasets, such as those in professional domains?
2.	Could future work incorporate real-time memorization detection to dynamically assess memorization levels in longer contexts?
3.	How does ICL performance vary between tasks that rely heavily on memorized information (e.g., factual knowledge) versus tasks that require more abstract reasoning or generalization?
4.	Is there a threshold where memorization starts to negatively impact performance, particularly if memorized information conflicts with task requirements?
5.	Would alternative architectures that prioritize generalization over memorization yield lower memorization rates in ICL while maintaining strong performance?
6.	How might the findings differ if fine-tuned LLMs with domain-specific training data are tested, where memorization may play a more prominent role?

---

> ### Author Response · Authors · 2024-11-23
> **Official Response by Authors (1/4)**
>
> Thank you for recognizing the novelty, detailed approach, and clear presentation of our work. We also appreciate your thoughtful and detailed feedback on our paper.
>
> We are grateful for the opportunity to address your concerns and answer your questions. Below, our responses are provided in the same order as your comments and questions.
>
> > 1. While the experiments are thorough, they are conducted in relatively simple datasets, limiting the paper’s ability to generalize findings to more complex, real-world tasks (e.g., legal, medical datasets).
>
> Thank you for your feedback. As outlined in Subsection 3.6, our dataset selection was guided by specific criteria: datasets needed to be part of the model's training data, labeled, possess a complex label space, and contain samples of a few dozen words/tokens in order to avoid reaching the maximum context length in many-shot scenarios. Given the limited transparency around the training data for both open-weight and closed-weight LLMs, we conducted a comprehensive investigation to identify datasets meeting all these criteria, leading to our selection of the four datasets used in this study.
>
> We acknowledge the value of incorporating datasets from diverse domains, including medical and legal fields. However, these datasets often contain lengthy samples that could conflict with our criterion of having samples limited to a few dozen words/tokens, to ensure we stay within the maximum context length in many-shot regimes.
>
> _Additionally, we want to highlight that the domain and complexity of downstream tasks are unlikely to significantly influence our findings. This is because our dataset selection was not driven by any specific domain, and our proposed approach focuses on measuring memorization through a text completion task, which is independent of the complexity of downstream tasks._
>
> > 2. The study does not address potential challenges in handling longer contexts, which are often needed in real-world applications and may limit the practicality of the proposed memorization detection method in large-scale LLMs.
>
> Thank you for your input. **We would like to clarify that our method is not limited by longer context samples.** Our decision to focus on datasets with samples containing a few dozen words/tokens was driven by our goal of thoroughly quantifying memorization across different ICL regimes, from zero-shot to many-shot. In fact, if someone wishes to apply our method to longer contexts, they can simply do so by including a few demonstrations (e.g., 25 demonstrations). This is because, according to our findings, most memorization becomes evident as soon as adding only a few demonstrations to the input prompt.
>
> **Furthermore, it is important to highlight that detecting memorization in longer contexts is actually easier than in shorter ones, known as the “discoverability phenomenon” (Carlini et al., 2023).** This is further supported by the findings of Carlini et al. (2023), who found that one key factor significantly increasing memorization in language models is the number of tokens of context used to prompt the model. In fact, conditioning the model on more tokens of context is more specific than conditioning the model on fewer tokens, naturally leading the model to assign a higher probability to the training data under these conditions.
>
> **Therefore, our approach is well-suited for both short and long-context documents, and our use of shorter samples was driven by our research questions aimed at studying memorization across various ICL regimes and was not due to any limitations of the proposed method itself.**

---

> ### Author Response · Authors · 2024-11-23
> **Official Response by Authors (2/4)**
>
> > 3. While the paper successfully identifies memorization as a factor, it does not provide a concrete analysis of how much ICL performance improvement is due to memorization versus actual generalization, leaving this as an open question.
>
> Thank you for your feedback. To analyze how much of the improvement in ICL performance is due to memorization versus other factors, such as generalization, we use the *coefficient of determination* ($R^2$). Specifically, $R^2$ allows us to measure the proportion of variance in ICL performance improvement that can be explained by memorization.
>
> The table below presents these results. As shown, in datasets where ICL leads to performance improvement (e.g., WNLI, TREC, and DBpedia), a significant portion of the improvement can be attributed to memorization. However, this does not hold in the case where ICL does not outperform zero-shot learning. For example, in the WNLI dataset under the segment pairs and label setting, 99.50% of the performance improvement is attributed to memorization, indicating that the improvement is almost entirely driven by memorization.
>
> P.S. For this analysis, we used memorization levels that were quantified using both *exact and near-exact matches.*
>
> | Setting             | WNLI  | TREC  | DBpedia | RTE   |
> |---------------------|-------|-------|---------|-------|
> | Full Information    | 0.969 | 0.804 | 0.824   | 0.305 |
> | Seg. Pairs & Labels  | 0.995 | 0.646 | 0.780   | 0.093 |
> | Only Seg. Pairs     | 0.980 | 0.708 | 0.789   | 0.089 |
>
> > 4. The experiments rely solely on GPT-4, limiting the generalizability of the findings to other LLMs. The authors could strengthen their conclusions by evaluating memorization across a range of models with varying training data and architectures.
>
> Thank you for your suggestion. We agree that adding more models would enhance our insights. However, we would like to clarify why this option was not feasible for us due to the high cost of our experiments.
>
> First, we did not have sufficient GPU resources to conduct experiments with long-content input prompts. As a result, we focused on proprietary LLMs where GPU power is supported by the model provider. As mentioned in Section 4 under Model, we conducted a pilot study with existing proprietary LLMs and selected GPTs based on the results. Initially, we aimed to control costs by using GPT-3.5. However, its 16k context limit prevented us from conducting experiments in many-shot regimes.
>
> To ensure comprehensive experiments across various regimes (i.e., zero-shot, few-shot, and many-shot), we managed our budget to use GPT-4, which offers a 32k context length, enabling many-shot experimentation. However, these experiments were quite expensive, making it infeasible to test additional closed- or open-weight LLMs. To give you an idea of the costs involved, OpenAI charges for API usage based on both **input** and **output** tokens. While the output tokens in our experiments were negligible, the input tokens were the primary cost driver. For example, in a 200-shot scenario, the input tokens ranged from 15k to 22k. OpenAI charges 60 dollars per 1 million input tokens, meaning each API call costs approximately 1 dollar in a 200-shot scenario. _For every data point in the left-hand plots of Figure 2, we tested 200 instances, resulting in a cost of 200 dollars per data point for 200-shot experiments._ Considering that *we repeated these experiments across four datasets and three different settings (full information, segment pairs and labels, and only segments),* the experiments became very expensive. In fact, such an undertaking is very difficult without industry-level resources.
>
> Finally, we hope this explanation justifies the budget constraints that led to our use of GPT-4 in our experiments.
>
> __Questions:__
>
> > 1. How would the memorization-performance correlation change if tested on larger, more diverse datasets, such as those in professional domains?
>
> Thank you for your question. While the memorization-performance correlation is inherently tied to the specifics of the training data, it is challenging to make definitive claims about other datasets without concrete evidence. However, for memorization to occur, it is necessary (though not sufficient) for the corresponding datasets to be part of the model’s training data. Larger and more diverse datasets, especially in professional domains, are more likely to overlap with the data available on the internet and, consequently, the training data. This increased overlap could enhance memorization, making the memorization-performance correlation more pronounced compared to smaller, less diverse datasets.

---

> ### Author Response · Authors · 2024-11-23
> **Official Response by Authors (3/4)**
>
> > 2. Could future work incorporate real-time memorization detection to dynamically assess memorization levels in longer contexts?
>
> Thank you for your insightful question. Yes, we do believe that future work can indeed build upon our proposed method for real-time memorization detection. There are two main reasons supporting this. First, our method is both straightforward and effective, scaling across different k-shot ICL regimes similarly to standard ICL. Second, as noted in our earlier response to item 2 above, our method is task/domain-agnostic and capable of handling both short and long contexts without limitations on sample length. Therefore, these flexibilities make it well-suited for dynamic assessment of memorization.
>
> > 3. How does ICL performance vary between tasks that rely heavily on memorized information (e.g., factual knowledge) versus tasks that require more abstract reasoning or generalization?
>
> Thank you for your thoughtful question. We would like to argue that our findings suggest two potential scenarios regarding ICL performance in this case.
>
> In the first scenario, if the factual accuracy of the memorized information from the training data aligns with the factual requirements of the downstream task, ICL can boost performance. This is because, as shown in our findings, ICL significantly increases the memorization level, and when this memorized information aligns with the factual knowledge needed for the downstream task, performance naturally improves.
>
> In the second scenario, if the factual accuracy of the memorized information from the training data conflicts with the factuality required for the downstream task, performance may initially decrease within a certain k-shot regime. However, as k increases, the model can overcome the memorized biases by conditioning on a substantial amount of new input information. Then, beyond this k-shot regime, performance can improve. Agarwal et al. (2024) illustrate this scenario in their paper in Figure 10. In fact, the findings of our work can now explain the underlying reason for this phenomenon. Also, it is also worth noting that the ability to overcome biases through ICL largely depends on factors such as the demonstrations used, the ICL capabilities of the model, the model size, and the severity of the biases present in the training data.
>
> In contrast, for tasks that require more abstract reasoning, a decline in ICL performance is less common. This is because LLMs often encounter similar reasoning tasks during their training, and the reasoning steps typically remain unaffected by changes in factual knowledge over time. For instance, converting speed from kilometers per hour to miles per hour relies on a fixed ratio and is not influenced by changes in factual data. Thus, in such cases, a decline in ICL performance generally occurs when the format or complexity of the downstream reasoning task significantly deviates from what the model has seen during its training.
>
> > 4. Is there a threshold where memorization starts to negatively impact performance, particularly if memorized information conflicts with task requirements?
>
> Thank you for your question. Yes, there is a threshold where memorization can negatively affect performance, especially when memorized information conflicts with the task requirements. To illustrate this, we would like to refer you again to Figure 10 in the work of Agarwal et al. (2024), which shows a case where original labels are flipped and the model is instructed to predict these flipped labels based on demonstrations with flipped labels. In this scenario, performance initially declines up to a certain number of demonstrations and then starts to improve as the number of demonstrations increases. In fact, this is a point where in-context information overcomes memorized information, leading to improved performance beyond this point.
>
> Also, it is important to note that this threshold varies across different models and tasks. As noted in our response to Question 3, factors such as demonstrations given, the model’s in-context capabilities, training data, and model size all influence this threshold.

---

> > ### Author Response · Authors · 2024-11-23
> > **Official Response by Authors (4/4)**
> >
> > > 5. Would alternative architectures that prioritize generalization over memorization yield lower memorization rates in ICL while maintaining strong performance?
> >
> > Thank you for your question. In fact, it is challenging to provide a definitive answer to this question, and there are two key reasons for this. First, we currently lack architectures that consistently prioritize generalization over memorization across all scenarios, including ICL. Second, the inner mechanics of ICL itself are not yet fully understood. Therefore, given the current state of knowledge, it is impossible to provide a universally accurate response. However, if we assume the existence of such architectures that **consistently** prioritize generalization over memorization across **all scenarios, including ICL,** it would be reasonable to argue that such architectures could indeed lower memorization rates in ICL while maintaining strong performance.
> >
> > > 6. How might the findings differ if fine-tuned LLMs with domain-specific training data are tested, where memorization may play a more prominent role?
> >
> > Thank you for your question. While our method remains effective in such cases, we believe the memorization-performance correlation observed in our findings would be more easily detectable, with memorization potentially reaching high levels, similar to what we observed with the WNLI dataset. This can be explained from several perspectives. First, memorization tends to increase significantly as the number of duplicated samples in the training data grows (Carlini et al., 2023), and fine-tuning can greatly contribute to this effect using only a few epochs. Second, when language models are fine-tuned on a highly specific domain, they are more prone to memorizing dataset peculiars (Carlini et al., 2023), which can further elevate the level of memorization surfaced by ICL.
> >
> > Regarding performance, since memorization is expected to play a stronger role due to the factors discussed, it could be more sensitive to the selection of demonstrations.
> >
> > ---
> >
> > Lastly, we hope our responses have effectively addressed your concerns. If you have any further questions or need additional clarification, please let us know. We would be more than happy to discuss any aspect in greater detail.
> >
> > ---
> >
> > References:
> >
> > [1] Quantifying Memorization Across Neural Language Models (Carlini et al., ICLR 2023)
> >
> > [2] Many-Shot In-Context Learning (Agarwal et al., NeurIPS 2024)

---

> ### Author Response · Authors · 2024-12-01
> **Kind Reminder About Our Rebuttals**
>
> Dear Reviewer QHTo,
>
> Thank you very much for taking the time to review our paper. As the rebuttal period is coming to an end, _we would like to kindly remind you that we have carefully and thoroughly responded to all your questions and concerns, and provided further analysis to determine how much ICL performance improvement is due to memorization versus actual generalization._ We deeply value your feedback and would greatly appreciate it if you could provide any additional input or confirm whether our responses have addressed your concerns.
>
> Thank you for your time and attention.
>
> Best regards,
> The Authors

---

### Official Review · Reviewer_AYex · 2024-11-10

**Soundness:** 3
**Presentation:** 2
**Contribution:** 2
**Rating:** 3
**Confidence:** 3

**Summary:**

The paper investigates the role of memorization in in-context learning (ICL) for LLMs. It shows that ICL significantly surfaces memorized training data, particularly when demonstrations without labels are used, and it establishes a strong correlation between memorization and improved model performance. The study finds that memorization levels increase with more demonstrations, with surfaced memorization reaching up to 75% in many-shot settings. Moreover, performance on memorized instances consistently surpasses that on non-memorized instances, raising questions about how much of ICL's success is due to true learning versus memorization.

**Strengths:**

This paper proposes a novel approach to quantify the level of memorization within ICL, contributing to a study in understanding the behavior of LLMs.

**Weaknesses:**

1. The paper presents findings that are already well-known in the field of machine learning: models perform better when they memorize training data, which is often referred to as "data leakage." It is unclear whether the primary motivation of this study is to investigate the relationship between ICL and memorization or to propose a method for quantifying memorization.

1. The paper claims to explore the "correlation between memorization and performance on downstream tasks" (lines 13-14). However, the Pearson correlation coefficient appears to be computed based on match scores across different k-shot settings. The memorization and downstream ICL tasks have different formats and purposes, which makes the correlation analysis questionable. Additionally, the negative correlation for the RTE dataset contrasts with the positive correlations for other benchmarks, casting doubt on the overall claim of correlation.

1. The authors do not provide an explanation for the observed experimental results. For example, the six observations drawn from Figure 2 are stated as mere observations without discussing the underlying reasons or contributing factors behind the results.

1. The comparison of different k values in evaluating memorization in Figure 2 is not adequately justified. The importance of the number of demonstrations as a variable for quantifying memorization remains unclear.

1. The use of 25-shot and 50-shot as examples of few-shot scenarios is too much. I expect that few-shot settings refer to 3-shot to 5-shot examples. Zero-shot results do not clearly indicate memorization level, as exact match criteria might be influenced by formatting issues rather than actual memorization. True few-shot settings, such as 3-shot, could provide a better output by constraining the output format, allowing for more precise evaluation.

1. Only GPT-4 is evaluated in the experiments. Including more LLMs in the experiments would provide stronger support for the paper's claims and offer broader insights.

1. The paper's presentation could be improved in several areas:
    * The repeated use of the phrase "memorization in ICL" is ambiguous, as it could imply that LLMs are memorizing the demonstrations instead of the training data. Since the study aims to explore memorization in LLMs using ICL as a tool, it would be clearer to use "memorization by ICL."
    * Figure 2 is intended to compare different settings, but the use of multiple curves for various datasets in a single subfigure makes it challenging to interpret comparisons. The authors should consider placing curves for different settings but the same dataset in each subfigure for clarity.

**Questions:**

1. How are non-memorized and memorized instances determined in Figure 3?
1. What is the relationship between the number of shots (k) in downstream ICL and the number of shots used in memorization experiments? The two tasks seem to serve different tasks with different purposes.

---

> ### Author Response · Authors · 2024-11-23
> **Official Response by Authors (1/4)**
>
> Thank you for recognizing the novelty of our approach. We are also grateful for your detailed review of our work.
>
> We appreciate the opportunity to respond to your concerns. In the following, we address them in the order they were presented.
>
> > 1. The paper presents findings that are already well-known in the field of machine learning: models perform better when they memorize training data, which is often referred to as "data leakage." It is unclear whether the primary motivation of this study is to investigate the relationship between ICL and memorization or to propose a method for quantifying memorization.
>
> Thank you for your feedback. It appears there may have been a misunderstanding. We appreciate the opportunity to clarify the primary motivation and key findings of our work.
>
> We would like to clarify that our study **does not focus** on data leakage/contamination, as **these topics were not discussed within the paper.** Additionally, the assertion that *"The paper presents findings that are already well-known in the field of machine learning: models perform better when they memorize training data."* **is not entirely consistent with our findings.** Specifically, our findings challenge the notion that memorization *always* leads to better performance. In our experiments with the RTE dataset, while memorization was observed, in-context learning (ICL) underperformed compared to zero-shot learning. Thus, models **do not** universally perform better when they exhibit memorization. As highlighted among our key findings, ICL improves performance when the surfaced memorization reaches a high level (approximately 40%); otherwise, it can lead to inferior performance compared to zero-shot learning. Furthermore, the “data leakage” scenario you mentioned pertains to our experimental design, which required the use of datasets present in the model’s training data to enable detection and quantification of memorization. Otherwise, it would have been unreasonable to detect/quantify memorization for data that was not part of the training.
>
> Regarding the primary motivation of our study, our objective is twofold: (1) to reveal memorization as a hidden factor influencing ICL and (2) to explore the relationship between this memorization and model performance. This motivation is articulated in lines 36-37 of the Introduction section. To achieve these objectives, we proposed a novel method to detect and quantify memorization that is surfaced by ICL. Without this method, identifying memorization as a factor impacting ICL would not have been feasible. Thus, quantifying memorization became a critical step for us to investigate its correlation with performance. **In summary, the two motivations you mentioned above are indeed interconnected, with our approach to detecting memorization serving as the foundation for quantifying memorization surfaced by ICL and examining its correlation with performance.**
>
> On the topic of novelty, we would like to emphasize the unique contributions of our study. **First, we would like to state that we are the first to identify memorization as a factor affecting ICL.** To the best of our knowledge, no prior research has highlighted the potential for ICL to increase memorization levels behind the scenes. **Second, we proposed a novel methodology to both *detect* and *quantify* memorization, scalable as simple as standard ICL by increasing demonstrations.** **Third, for the first time, we measured the correlation between this surfaced memorization and improved performance in ICL.** With this in mind, if there is existing literature discussing similar findings related to ICL memorization, we would appreciate any references you could share, as it would allow us to compare our work and underscore our contributions more effectively.

---

> ### Author Response · Authors · 2024-11-23
> **Official Response by Authors (2/4)**
>
> > 2. The paper claims to explore the "correlation between memorization and performance on downstream tasks" (lines 13-14). However, the Pearson correlation coefficient appears to be computed based on match scores across different k-shot settings. The memorization and downstream ICL tasks have different formats and purposes, which makes the correlation analysis questionable. Additionally, the negative correlation for the RTE dataset contrasts with the positive correlations for other benchmarks, casting doubt on the overall claim of correlation.
>
> Thank you for your input. _However, we would like to point out that the precise phrasing in lines 13-14 of our paper is: “This study is the first to show how ICL surfaces memorized training data and to explore the correlation between this memorization and performance on downstream tasks **across various ICL regimes: zero-shot, few-shot, and many-shot.**”_ As stated, the objective is to investigate the relationship between memorization and performance **across different ICL regimes, ranging from zero-shot to many-shot.** Therefore, varying k in the k-shot regimes is fundamental and aligns with our research objectives. Otherwise, limiting the analysis to only two static data points—one representing memorization and the other representing performance—would render the Pearson correlation calculation statistically meaningless.
>
> Furthermore, we would like to emphasize that **both memorization and downstream ICL tasks maintain the same format, as shown in Figure 1.** In fact, one of the key strengths of our proposed methodology lies in its ability to scale the number of demonstrations in a manner identical to standard ICL. Therefore, they are the same in terms of format. Additionally, both tasks share the same purpose: improving performance on downstream target variables (y). However, since they are distinct tasks, it is natural that they have different target variables.
>
> **Therefore, based on the points discussed in the two above paragraphs, we conclude that our correlation analysis is sound and well-supported.**
>
> Regarding the negative correlation for the RTE dataset, we would like to emphasize that, as discussed thoroughly in various sections of our paper (lines 17-18, lines 102-103, and lines 434-436), when memorization surfaced by ICL reaches a substantial level, it can positively impact performance. In fact, this finding is one of the key outcomes of our study. Specifically, for the RTE dataset, the level of memorization is not substantial when ICL is applied, which results in performance that is lower than that of zero-shot learning. _Additionally, we have consistently included a **conditional statement** in our paper whenever referring to improved performance by ICL and its correlation with surfaced memorization._ **Therefore, we have not made a blanket claim about the correlation between memorization and performance.**
>
> > 3. The authors do not provide an explanation for the observed experimental results. For example, the six observations drawn from Figure 2 are stated as mere observations without discussing the underlying reasons or contributing factors behind the results.
>
> Thanks for the suggestion. We would like to emphasize that the challenge of explaining the underlying reasons for observed experimental results is a widely acknowledged limitation in the field of Deep Learning, especially concerning large and deep models. _In fact, this is not specific to our work; it extends to many studies in this area, given the limited theoretical foundation currently available._ As such, much of the research, including ours, is necessarily practical in nature.
>
> For example, in the relevant and highly regarded work of Carlini et al. (2023) on quantifying memorization in language models, no definitive reasons are provided regarding why memorization occurs or why there is a log-leaner relationship between memorization and other factors, such as model size. Instead, such studies present practical observations aimed at shedding light on the behavior of models, without delving into causal explanations.
>
> Moreover, the inherent complexity of current models, the opaque nature of deep learning processes, and limited transparency concerning the data and training paradigms add further layers of difficulty to understanding the underlying reasons behind these phenomena.
>
> _In our case, providing reasons for the observed findings would necessitate a precise understanding of why memorization occurs in language models and the exact mechanisms governing ICL. To the best of our knowledge, no existing works have conclusively established such explanations._ **Therefore, we are constrained by the current state of knowledge in providing definitive underlying reasons for our observed findings.**

---

> ### Author Response · Authors · 2024-11-23
> **Official Response by Authors (3/4)**
>
> > 4. The comparison of different k values in evaluating memorization in Figure 2 is not adequately justified. The importance of the number of demonstrations as a variable for quantifying memorization remains unclear.
>
> Thanks for your feedback. We would like to clarify that we have justified the use of different k values for evaluating memorization *multiple times* throughout the paper. Specifically, **Subsection 3.4 has been dedicated to this topic.** Additionally, we revisited this in two other places: **lines 12-14 of the Abstract** and **lines 52-77 of the Introduction.** Moreover, we have discussed the importance of the number of demonstrations as a variable for quantifying memorization in all the aforementioned sections/lines. In a nutshell, defining the number of demonstrations as a variable is essential for our experiments, as studying memorization across different ICL regimes would not be possible otherwise.
>
> > 5. The use of 25-shot and 50-shot as examples of few-shot scenarios is too much. I expect that few-shot settings refer to 3-shot to 5-shot examples. Zero-shot results do not clearly indicate memorization level, as exact match criteria might be influenced by formatting issues rather than actual memorization. True few-shot settings, such as 3-shot, could provide a better output by constraining the output format, allowing for more precise evaluation.
>
> Thank you for your feedback. We would like to point out that there is no definitive number of shots that defines a few-shot versus a many-shot regime. However, with modern long-context LLMs capable of processing a few thousand to a few million tokens as input, these numbers can indeed be considered few-shot. Additionally, recent shifts towards many-shot paradigms (Agarwal et al., 2024; Bersch et al., 2024) and increased test-time computation (OpenAI o1) suggest that these numbers effectively reflect the current paradigm in the field.
>
> As for the results from zero-shot learning, it is important to note that **all experiments** in the work of Golchin and Surdeanu (2024), whose method we adapted, were *successfully* conducted under **zero-shot learning.** _Upon further review of this paper, we did not find any reported formatting issues influencing quantifying memorization, nor did we encounter such issues in our own experiments._
>
> Further, as mentioned previously, in this study, our objective is to examine memorization across various regimes, from zero-shot to many-shot. Therefore, experimenting with zero-shot scenarios was necessary to address the research questions posed in our study effectively.
>
> > 6. Only GPT-4 is evaluated in the experiments. Including more LLMs in the experiments would provide stronger support for the paper's claims and offer broader insights.
>
> Thank you for your suggestion. We agree that adding more models would enhance our insights. However, we would like to clarify why this option was not feasible for us due to the high cost of our experiments.
>
> First, we did not have sufficient GPU resources to conduct experiments with long-content input prompts. As a result, we focused on proprietary LLMs where GPU power is supported by the model provider. As mentioned in Section 4 under Model, we conducted a pilot study with existing proprietary LLMs and selected GPTs based on the results. Initially, we aimed to control costs by using GPT-3.5. However, its 16k context limit prevented us from conducting experiments in many-shot regimes.
>
> To ensure comprehensive experiments across various regimes (i.e., zero-shot, few-shot, and many-shot), we managed our budget to use GPT-4, which offers a 32k context length, enabling many-shot experimentation. However, these experiments were quite expensive, making it infeasible to test additional closed- or open-weight LLMs. To give you an idea of the costs involved, OpenAI charges for API usage based on both **input** and **output** tokens. While the output tokens in our experiments were negligible, the input tokens were the primary cost driver. For example, in a 200-shot scenario, the input tokens ranged from 15k to 22k. OpenAI charges 60 dollars per 1 million input tokens, meaning each API call costs approximately 1 dollar in a 200-shot scenario. _For every data point in the left-hand plots of Figure 2, we tested 200 instances, resulting in a cost of 200 dollars per data point for 200-shot experiments._ Considering that *we repeated these experiments across four datasets and three different settings (full information, segment pairs and labels, and only segments),* the experiments became very expensive. In fact, such an undertaking is very difficult without industry-level resources.
>
> Finally, we hope this explanation justifies the budget constraints that led to our use of GPT-4 in our experiments.

---

> ### Author Response · Authors · 2024-11-23
> **Official Response by Authors (4/4)**
>
> > 7. The paper's presentation could be improved in several areas:
> > - The repeated use of the phrase "memorization in ICL" is ambiguous, as it could imply that LLMs are memorizing the demonstrations instead of the training data. Since the study aims to explore memorization in LLMs using ICL as a tool, it would be clearer to use "memorization by ICL."
>
> Thanks! We revised our wording from "memorization in ICL" to "memorization by ICL" where applicable. Modifications are indicated in blue in the current revised version of the paper.
>
> > - Figure 2 is intended to compare different settings, but the use of multiple curves for various datasets in a single subfigure makes it challenging to interpret comparisons. The authors should consider placing curves for different settings but the same dataset in each subfigure for clarity.
>
> Thanks for the suggestion. However, we would like to clarify that we already have these subfigures in Figure 3, where curves from different settings for each dataset are plotted within a single subfigure.
>
> __Questions:__
> > 1. How are non-memorized and memorized instances determined in Figure 3?
>
> Thank you for the question. **As detailed in Subsection 3.3,** we identify memorized and non-memorized instances using the definitions provided for exact, near-exact, and inexact matches in Subsection 3.1. Specifically, we consider exact and near-exact matches as memorized instances and inexact matches as non-memorized instances.
>
> > 2. What is the relationship between the number of shots (k) in downstream ICL and the number of shots used in memorization experiments? The two tasks seem to serve different tasks with different purposes.
>
> Thank you for your question. The number of shots in downstream ICL and in memorization experiments is the **same.** As previously elaborated in item 2 above, **both tasks share the same purpose: improving performance on downstream target variables (y).** However, since they are distinct tasks, it is natural that they have different target variables.
>
> ---
>
> Finally, we hope our responses have effectively addressed your concerns. We would also be happy to discuss any points in further detail if you remain unconvinced by our answers.
>
> ---
>
> References:
>
> [1] Quantifying Memorization Across Neural Language Models (Carlini et al., ICLR 2023)
>
> [2] Many-Shot In-Context Learning (Agarwal et al., NeurIPS 2024)
>
> [3] In-Context Learning with Long-Context Models: An In-Depth Exploration (Bersch et al., arXiv 2024)
>
> [4] Time Travel in LLMs: Tracing Data Contamination in Large Language Models (Golchin and Surdeanu, ICLR 2024)

---

> > ### Comment · Reviewer_AYex · 2024-11-25
> >
> > Thank you to the authors for providing responses to my previous comments. I have some significant concerns regarding this paper.
> >
> > 1. Insufficient Experimental Support for Claimed Findings
> >
> >     The experimental results do not sufficiently support the primary claims of this paper. According to the authors, the main finding is that memorization does not always lead to better performance, and that ICL improves performance when the surfaced memorization reaches a high level (approximately 40%). However, these findings are supported by only **one dataset** (RTE) and **one LLM** (GPT-4).
> >
> >     This narrow experimental scope raises concerns about the generalizability of the conclusions. For instance, the observed behavior in RTE could be coincidental, driven by dataset-specific characteristics such as its size, class distribution, or the unknown distribution of GPT-4’s training data. Notably, in the RTE experiments, downstream ICL performance decreases as the $k$ increases, which could result from factors unrelated to memorization.
> >     I strongly recommend that the authors include additional datasets and LLMs in their experiments. I understand that the cost may be high, but without broader empirical evidence, the conclusions remain speculative and lack robust support.
> >
> >     Furthermore, the analysis of "Performance for Memorized Instances" (Figure 3, left column) suggests that memorization may act as a **random factor** in ICL performance. The trends of "Performance for Memorized Instances" in the figure are contradictory: while WNLI and RTE show ICL outperforming zero-shot on memorized instances, TREC and DBpedia show the opposite. These inconsistencies imply that high memorization level (100%) has no reliable relationship to downstream ICL performance, further undermining the central claim.
> >
> > 1. Non-Standard Evaluation on Train Split for ICL Performance
> >
> >     The evaluation of downstream ICL performance on the **train split** of a benchmark is unconventional and problematic. In typical ICL research, performance is evaluated on the test split to assess generalization and to fairly compare with baselines. This paper, however, evaluates ICL on the train split, which has been verified as part of GPT-4's training data, introducing **data leakage**. Although the authors clarify in Section 3.5 and 3.6 that their study investigates ICL and memorization under a data-leakage scenario, the findings derived from data-leakage conditions do not generalize to conventional ICL where generalization is critical. Therefore, the technical contributions to practical ICL are limited and the results are not directly applicable to improving ICL.
> >
> > 1. Misuse of Pearson Correlation Coefficient
> >
> >     When calculating the Pearson coefficient, the paper matches $k_1$-shot memorization scores to $k_2$-shot ICL performance scores, assuming $k_1 = k_2$. However, this assumption is problematic because memorization and ICL are distinct tasks with different goals and characteristics: memorization measures the ability to recall training data, while ICL performance assesses downstream task capabilities. There are other possible mappings. For example, $k_1 = 10 \times k_2$ can be analyzed because memorization tasks may require more demonstrations to converge, whereas ICL tasks need fewer. Other mapping $k_1=200-k_2$, matching many-shot to zero-shot, is also reasonable.
> >
> >     The choice of $k_1 = k_2$ seems **coincidental** without additional justification. For example, it is unclear how the 25-shot memorization task directly impacts the 25-shot ICL task. I suggest the authors clarify the rationale behind this choice and explore whether alternative mappings would yield similar or different conclusions.
> >
> > Minor Issues for Clarification
> >
> > 1. > ... downstream ICL tasks maintain the same format, as shown in Figure 1
> >
> >     The specific format of ICL tasks is not clearly illustrated in Figure 1 or the Appendix.
> >
> > 2. Memorized vs non-memorized instances
> >
> >     Subsection 3.3 lacks details about the criteria used to define exact and near-exact matches for memorization, specifically the value of $k$. Additionally, the paper does not report how many instances in each dataset are categorized as memorized vs. non-memorized.

---

> ### Author Response · Authors · 2024-11-25
> **Official Response by Authors (1/2)**
>
> Thank you for your questions and engagement during the rebuttal period. We are happy to answer your questions and address your concerns. Please find our responses below.
>
> > 1. Insufficient Experimental Support for Claimed Findings
>
> Thank you for your feedback. We would like to clarify that the key finding—_memorization does not always lead to better performance, and ICL improves performance when memorization reaches a high level (approximately 40%)_—__is supported by three datasets, not just one.__ Specifically, this __conclusion is based on the results from the TREC, WNLI, and DBpedia datasets, rather than the RTE dataset.__
>
> Additionally, if the behavior observed in the RTE dataset were purely coincidental, it would likely have occurred in only one or two ICL scenarios. __However, this pattern appeared consistently across all ICL regimes when k = {25, 50, 100, 200}.__ This consistency indicates that the behavior is not coincidental. _Moreover, it cannot be attributed to dataset size or class distribution since we uniformly subsampled 200 instances from each dataset, evenly balanced across labels, as detailed in lines 288–291 in Section 4 under Data. Similarly, it cannot be explained by the unknown distribution of GPT-4's training data, as the tasks we used were similar, well-known, and frequently encountered during LLM training._
>
> As mentioned previously, while we wish we could conduct further experiments, the associated costs make this infeasible. Nonetheless, our findings are supported by three datasets across three distinct regimes (zero-shot, few-shot, and many-shot) and three experimental settings (full information, segment pairs and labels, and segment pairs only). We believe these experiments are sufficient to substantiate our conclusions.
>
> Regarding your suggestion that memorization may act as a random factor in ICL performance, we respectfully disagree with this interpretation. This is because, as the number of demonstrations increases, more memorized instances are included, and performance is averaged over a larger number of memorized examples. Therefore, the resulting performance trend for memorized instances is both natural and consistent. In fact, the correct interpretation is that the performance of memorized instances always exceeds the overall performance plot, consistently making a positive contribution to the overall performance. In contrast, non-memorized instances have a negative impact on the overall performance.
>
> > 2. Non-Standard Evaluation on Train Split for ICL Performance
>
> Thank you for your feedback. As emphasized throughout the paper (Subsections 3.5 and 3.6), investigating memorization for data not included in the model’s training is not reasonable. Therefore, we believe our experimental design is robust and appropriate for addressing the research questions we aim to study. __Moreover, whether we use the train or test set does not affect the conclusions we have drawn. If the model’s performance under ICL stems from generalization, memorization should remain low even when the dataset has been seen during training. However, our results show this is not the case.__
>
> To further demonstrate the generalization of our findings to conventional ICL, we refer to the work of Mirzadeh et al. (2024). _Their research examines the scenario you mentioned and concludes that LLMs do not generalize beyond their training data. Instead, LLMs replicate identical reasoning steps from their training data when facing new problems._ However, in such cases, quantifying memorization is not feasible because the reasoning steps are scattered and lack a structured format for memorization measurement.
>
> __As a result, based on both our findings and those of others, we argue that our results are valid and generalizable to conventional ICL scenarios.__
>
> > 3. Misuse of Pearson Correlation Coefficient
>
> Thank you for your input. However, we would like to clarify that __we do not assume $k_1 = k_2$; rather, they are explicitly equal, as detailed in lines 303-305 of Section 4 under Demonstrations.__ Specifically, in the $k_1$-shot scenario in standard ICL, $k_1$ demonstrations are included in the input prompt, and these same $k_1$ demonstrations are used in the scenarios where we measure memorization. Therefore, $k_1 = k_2$.
>
> Additionally, regarding your suggestion of using $k_1 = 10 \times k_2$ or $k_1 = 200 - k_2$, __neither of these suggestions correctly represents the amount of memorization surfaced in $k$-shot ICL.__ The proper approach to evaluating the relationship between surfaced memorization in the $k$-shot scenario and performance is to use the __same number and identical demonstrations,__ as we did in the paper and as noted in lines 303-305 of Section 4 under Demonstrations.

---

> > ### Author Response · Authors · 2024-11-25
> > **Official Response by Authors (2/2)**
> >
> > > The specific format of ICL tasks is not clearly illustrated in Figure 1 or the Appendix.
> >
> > Thank you for your feedback. We believe the community already has a clear understanding of what ICL looks like, which is why we did not mention it. However, we will include it in the final version of the paper.
> >
> > > Subsection 3.3 lacks details about the criteria used to define exact and near-exact matches for memorization, specifically the value of $k$.
> >
> > Thank you for your feedback. __We respectfully disagree with the assertion that Subsection 3.3 lacks sufficient details about the criteria for defining exact and near-exact matches.__ To clarify, here is the exact text from Subsection 3.3: *"__We compute the performance on the samples for which we assess memorization and analyze the correlation between performance and memorization across our three settings using the Pearson correlation (Pearson, 1895) [value of $k$].__ In addition, we separately evaluate performance for memorized and non-memorized instances across ICL regimes to further explore this relationship. __According to Subsection 3.1, instances that are replicated exactly or nearly exactly are considered memorized, while those replicated inexactly are considered non-memorized [criteria].__"* This explicitly outlines the criteria and the value of $k$.
> >
> >
> > > Additionally, the paper does not report how many instances in each dataset are categorized as memorized vs. non-memorized.
> >
> > Thank you for your feedback. __However, Figures 2 and 3 clearly illustrate this information using percentages.__ To determine the exact number of instances in each dataset categorized as memorized, you can simply multiply the percentage reported for each k-shot by 200, as all our experiments involved 200 instances.
> >
> >
> > ---
> >
> > Please let us know if you have any further questions or need additional clarification. We would be happy to discuss and provide more details.
> >
> > ---
> >
> > References:
> >
> > [1] GSM-Symbolic: Understanding the Limitations of Mathematical Reasoning in Large Language Models (Mirzadeh et al., arXiv 2024)

---

> ### Author Response · Authors · 2024-12-01
> **Kind Reminder About Our Rebuttals**
>
> Dear Reviewer AYex,
>
> Thank you very much for taking the time to review our paper. As the rebuttal period is coming to an end, _we would like to kindly remind you that we have carefully and thoroughly responded to all your questions and concerns._ We deeply value your feedback and would greatly appreciate it if you could provide any additional input or confirm whether our responses have addressed your concerns.
>
> Thank you for your time and attention.
>
> Best regards,
> The Authors

---

> > ### Comment · Area_Chair_kHB4 · 2024-12-04
> > **Concerns addressed?**
> >
> > Would be great if you can acknowledge whether author's rebuttal addressed your concerns.

---

### Author Response · Authors · 2024-12-02
**Kind Reminder About Our Rebuttals**

Dear Reviewers,

Thank you for reviewing our work and providing your valuable feedback. Since today marks the final day of the rebuttal-discussion period and our last opportunity to improve our submission based on your feedback, we would appreciate it if you could provide any additional input or confirm whether our responses have addressed your concerns.

Best regards,
The Authors

---

### Meta-Review · Area_Chair_kHB4 · 2024-12-19

**Metareview:**

This paper studies memorization in in-context learning and explores its correlation with downstream performance of LLMs.  The key findings suggest that ICL significantly surfaces memorization compared to zero-shot learning, and demonstrations without labels are the effective in doing so, and there's a strong correlation between performance and memorization in ICL when it outperforms zero-shot ICL. The paper is the first to systematically examine the relationship between ICL and memorization in LLMs, providing new insights into how memorized knowledge influences ICL performance. However, the paper has several shortcomings weaknesses:

- Limited generalizability due to reliance on a single LLM (GPT-4). Several reviewers pointed out that the study's conclusions might not generalize to other LLMs with different training data and different architectures, which I am also uncertain about. As such, the paper's findings might be an artifact of GPT-4, rather than a general finding. While authors pointed out computational or API cost, recent long context models such as Gemini Flash are substantially cheaper than GPT-4o. Another option could be smaller versions of open-weights long-context LLMs, such as LLama2-32K.

- Tasks with Conflicting results and Limited Analysis. The paper only studies 2 NLI and 2 classification tasks, where the negative correlation for the RTE dataset contrasts with the positive correlations for other benchmarks, casting doubt on the overall claim of correlation. Furthermore, the paper presents a list of observations, without running detailed ablations to further understand those results.

- Deliberate data leakage in experiments. The use of datasets that are part of GPT-4's training data introduces data leakage and limits the generalizability of the findings to conventional ICL scenarios where generalization is critical. It is possible that ICL operates mainly through memorization on such leaked datasets, while on conventional datasets ICL's mechanism might be different. As such, experiments to understand the role of memorization in ICL performance (possibly on synthetic datasets that can be controlled) seems critical to understand for this paper's relevance.

Overall, while the paper studies a timely and relevant topic, my recommendation is to reject the paper based on the weaknesses pointed above. That said, I encourage the authors to revise the paper for a stronger submission.

**Additional Comments On Reviewer Discussion:**

During the rebuttal period, the authors responded to the reviewers' concerns by clarifying their methodology, providing additional explanations, and adding new sections to the paper to address practical implications and further details on findings.  They also acknowledged the limitations of their study and the need for future work to address those limitations.  Despite these efforts, some reviewers remained unconvinced about the novelty and generalizability of the findings due to the limited experimental scope.  The authors' responses were considered in the final decision, but the narrow experimental scope made me lean towards rejecting the paper.

---

### Decision · Program_Chairs · 2025-01-22

Reject